# CHAT-UNIVI: A UNIFIED VISION-LANGUAGE MODEL FOR IMAGE AND VIDEO UNDERSTANDING

## ABSTRACT

Large language models have demonstrated impressive universal capabilities across a wide range of open-ended tasks and have extended their utility to encompass multimodal conversations. In this study, we introduce Chat-UniVi, a **Uni**fied **Vi**sion-language model capable of comprehending and engaging in conversations involving images and videos. Specifically, Chat-UniVi uniformly represents images and videos using a collection of dynamic visual tokens. This novel representation framework empowers the model to efficiently utilize a limited number of visual tokens to simultaneously capture the spatial details necessary for images and the comprehensive temporal relationship required for videos. Besides, we leverage a multi-scale representation that equips large language models to perceive both high-level semantic concepts and low-level visual details. More encouragingly, Chat-UniVi is trained on a mixed dataset containing both images and videos, making it directly applicable to tasks involving both mediums without the need for any modifications. Extensive experimental results demonstrate that Chat-UniVi, as a unified model, consistently surpasses even the existing methods exclusively designed for either images or videos. To the best of our knowledge, Chat-UniVi represents the first successful unified multimodal large language model that consistently outperforms both dedicated image and video models.

## 1 INTRODUCTION

Large language models (LLMs), e.g., GPT-3 (Brown et al., 2020), GPT-4 (OpenAI, 2023), and LLaMA (Touvron et al., 2023a;b), showcase substantial universal capabilities that pave the way for achieving general artificial intelligence. However, language represents just one facet of communication. Visual information serves to augment and enhance our comprehension of the world. Therefore, there exists a burgeoning interest in developing a multimodal conversation model that can accommodate various input modalities simultaneously, including images and videos.

Recent advances in multimodal conversation models, e.g., MiniGPT-4 (Zhu et al., 2023), MultiModal-GPT (Gong et al., 2023), and mPLUG-Owl (Ye et al., 2023), focus on integrating visual tokens into LLMs. Despite their commendable progress, existing methods often specialize in either image or video inputs. For instance, methods that prioritize image inputs, e.g., LLaVA (Liu et al., 2023), typically employ a larger number of visual tokens to attain finer spatial understanding. Conversely, methods concentrating on video inputs, e.g., Video-ChatGPT (Maaz et al., 2023), often compromise spatial comprehension per frame to accommodate more frames for modeling temporal relationships. Although some methods, e.g., Flamingo (Alayrac et al., 2022), can extract a fixed number of tokens for each image and video using a query transformer, their primary emphasis remains on image understanding, lacking the capability to effectively model temporal comprehension, thus resulting in a limited understanding of videos. Therefore, it is crucial and challenging to enable LLMs for both image and video comprehension within a unified framework.

In this paper, we introduce Chat-UniVi, a **Uni**fied **Vi**sion-language model designed to proficiently comprehend and engage in conversations about both images and videos. Chat-UniVi uniformly represents images and videos using a collection of dynamic visual tokens, enabling it to concurrently capture the spatial details of images and the comprehensive temporal relationship of videos. As illustrated in Fig. 1, images can be depicted through visual tokens of diverse sizes. For example, the primary object, i.e., the sheep in Fig. 1, necessitates a fine-grained representation with numerous

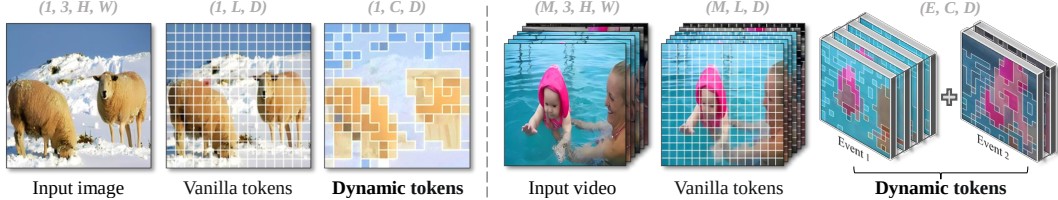

Figure 1: **The unified representation framework for images and videos utilizing dynamic visual tokens.** $H$ and $W$ represent the height and width of the input, respectively. $L$, $D$, $M$, $C$, and $E$ denote the number of vanilla visual tokens, the feature dimension, the frame length, the number of dynamic visual tokens, and the number of events, respectively.

visual tokens, while the background, i.e., the snow-capped mountain, can be sufficiently modeled with only one visual token. In the case of videos, the video is initially divided into several events, and subsequently, these visual tokens expand over frames within each event to encapsulate frame-level dynamics. Such unified representation for both images and videos significantly reduces the number of visual tokens while maintaining the expressive capabilities of the model. Moreover, longer videos are assigned more visual tokens and are therefore better suited for variable-length video understanding.

To obtain these dynamic visual tokens, we propose a parameter-free token merging method for progressively merging visual tokens with similar semantic meanings. Specifically, starting with visual tokens initialized by the Vision Transformer (Dosovitskiy et al., 2021), we gradually group them by applying the k-nearest-neighbor based density peaks clustering algorithm, i.e., DPC-KNN (Du et al., 2016), on the token features. When it comes to videos, we also utilize DPC-KNN on the frame features to get events. At each merging step, we merge the visual tokens assigned to the same cluster by averaging their token features. Finally, we supply a multi-scale representation to the LLMs. The upper layers of the multi-scale representation encompass high-level semantic concepts, while the lower layers emphasize visual details representations.

The proposed Chat-UniVi has two compelling advantages: **First**, its unified image and video modeling method allows training on the mixed dataset of image and video, enabling direct application to both image and video tasks without any modifications. **Second**, the multi-scale representation contributes to the comprehensive understanding of images and videos, empowering Chat-UniVi to adapt to various tasks, including employing high-level representation for semantic understanding and low-level representation for generating detailed descriptions. We evaluate Chat-UniVi on both image and video understanding tasks. Compared to other methods focused exclusively on either images or videos, Chat-UniVi consistently demonstrates superiority in comprehending images and videos. Moreover, we also provide evidence of the advantages of joint training of images and videos for multimodal large language models. The main contributions are summarized as follows:

- To the best of our knowledge, the proposed Chat-UniVi is the first successful unified vision-language model that consistently outperforms both dedicated image and video models.
- We uniformly represent images and videos using multi-scale dynamic visual tokens and propose a parameter-free token merging method to obtain these visual tokens.
- Without fine-tuning, Chat-UniVi attains competitive performance in both image and video tasks and achieves impressive results in the object hallucination benchmark.

## 2 RELATED WORK

**Large Language Models.** Recently, large language models (Kenton & Toutanova, 2019; Radford et al., 2019; Raffel et al., 2020; Vaswani et al., 2017) have made disruptive progress, primarily attributed to the expansion of training data and the substantial increase in model parameters. Inspired by the success of GPT-3 (Brown et al., 2020), numerous large language models have subsequently been developed, including PaLM (Chowdhery et al., 2022), OPT (Zhang et al., 2022), BLOOM (Scao et al., 2022), InstructGPT (Ouyang et al., 2022), and ChatGPT (OpenAI, 2022). However, language represents just one facet of communication. Visual information serves to augment and enhance our comprehension of the world (Labiosa et al.; Jin et al., 2022; 2023b). In this work, we introduce Chat-UniVi, designed to not only comprehend and generate responses from text but also incorporate visual inputs, thereby providing a more comprehensive and immersive context for response generation.

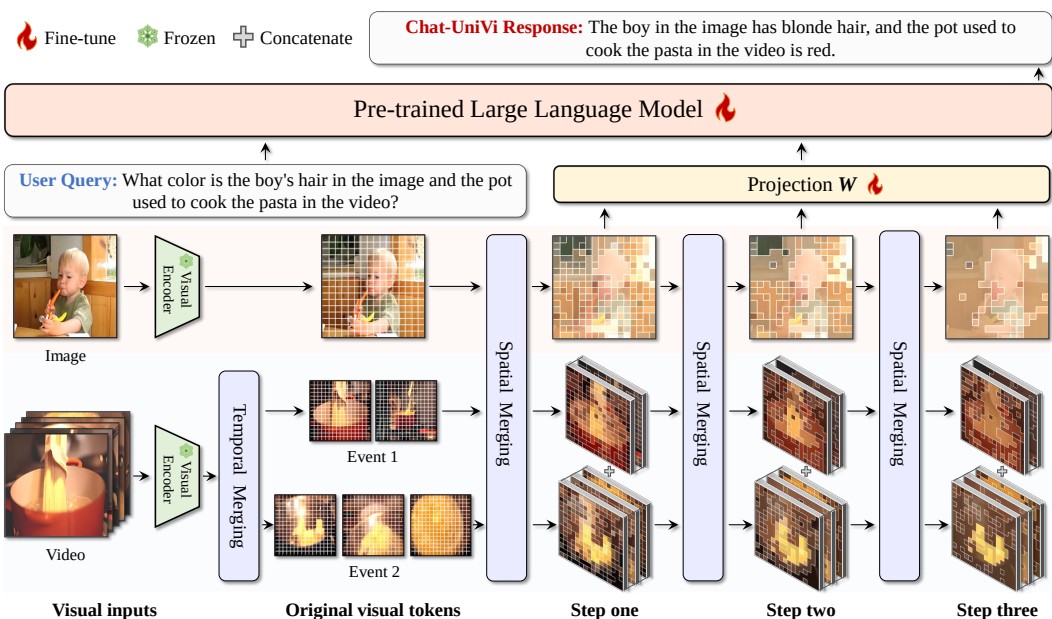

Figure 2: **The overview of the proposed Chat-UniVi for conversations containing both images and videos.** Chat-UniVi uniformly represents images and videos using a collection of dynamic visual tokens and provides a multi-scale representation that equips large language models to perceive both high-level semantic concepts and low-level visual details.

**Large-scale Multimodal Models.** Existing large-scale multimodal models can be broadly categorized into two classes. The first class of methods involves using LLMs as a dispatch scheduler, facilitating connections between various expert models to handle different vision tasks. These methods are exemplified by VisualChatGPT (Wu et al., 2023a), HuggingGPT (Shen et al., 2023), MM-REACT (Yang et al., 2023), and ViperGPT (Surís et al., 2023). The second class of methods emphasizes the integration of models from different modalities into end-to-end trainable models. Representatives of this approach include GPT-4 (OpenAI, 2023), Mini-GPT4 (Zhu et al., 2023), Flamingo (Alayrac et al., 2022), BLIP-2 (Li et al., 2023b), InstructBLIP (Dai et al., 2023), Otter (Li et al., 2023a), mPLUG-Owl (Ye et al., 2023), LLaMA-Adapter (Zhang et al., 2023b), and LLaMA-Adapter V2 (Gao et al., 2023). More recently, there have also been several dedicated multimodal models tailored for video processing, such as Video-ChatGPT (Maaz et al., 2023), VideoChat (Li et al., 2023c), and Video-LLaMA (Zhang et al., 2023a). Despite their commendable progress, existing methods often focus exclusively on either image or video inputs. In this work, we focus on developing an end-to-end trained multimodal model for both image and video tasks. Although Flamingo also supports both image and video inputs, it can only extract a fixed number of tokens for videos of varying lengths with a query transformer. Recent works (Wu et al., 2023b; Chen et al., 2023) have explored the use of separately pre-trained image and video encoders for processing, but these methods introduce model redundancy and prove challenging to train together. Hence, it does not align with our focus on achieving a unified vision-language model. In contrast to the previous works, Chat-UniVi uniformly represents images and videos using multi-scale dynamic visual tokens.

## 3 METHODOLOGY

Chat-UniVi aims to model images and videos concurrently within a language sequence that can be comprehended by Large Language Models (LLMs) in a unified framework. Chat-UniVi achieves this by uniformly representing images and videos through a set of dynamic visual tokens, bridging the intricate spatial details of images with the broader temporal comprehension needed for videos. The overview of the proposed Chat-UniVi is shown in Fig. 2.

### 3.1 DYNAMIC VISUAL TOKENS FOR IMAGE AND VIDEO

Building upon the foundation of the vanilla Vision Transformer, most methods generate equally important visual tokens by dividing the image into regular and fixed grids. However, it is evident

that not all regions hold equal significance in vision-language tasks. For example, capturing the background may require only a single visual token. Drawing inspiration from this insight, We amalgamate non-essential tokens to derive dynamic vision regions as input for LLMs.

**Spatial Visual Token Merging.** For an input image, we adopt the vision encoder of CLIP (ViT-L/14) (Radford et al., 2021) to provide the original visual tokens $\mathbb{Z} = \{z_i\}_{i=1}^{L}$, where $L$ is the number of visual tokens each image is divided into. To amalgamate non-essential visual tokens, we utilize DPC-KNN (Du et al., 2016), a k-nearest neighbor-based density peaks clustering algorithm, to cluster the visual tokens. Starting with visual tokens $\mathbb{Z} = \{z_i\}_{i=1}^{L}$ initialized by the vision transformer, we first compute the local density $\rho_i$ of each token $z_i$ according to its $K$-nearest neighbors:

$$\rho_i = \exp\Big(-\frac{1}{K}\sum_{z_k \in \text{KNN}(z_i,\mathbb{Z})} \|z_k - z_i\|^2\Big), \tag{1}$$

where $\text{KNN}(z_i, \mathbb{Z})$ is the $K$-nearest neighbors of $z_i$ in $\mathbb{Z}\backslash\{z_i\}$. "$\mathbb{Z}\backslash\{z_i\}$" denotes removing $\{z_i\}$ from $\mathbb{Z}$. Intuitively, $\rho_i$ denotes the local density of token $z_i$. Then, we compute the distance index $\delta_i$ of the token $z_i$, which is formulated as:

$$\delta_i = \begin{cases} \min_{j:\rho_j > \rho_i} \|z_j - z_i\|^2, & \text{if } \exists j \text{ s.t. } \rho_j > \rho_i. \\ \max_j \|z_j - z_i\|^2, & \text{otherwise.} \end{cases} \tag{2}$$

In essence, $\delta_i$ represents the distance between the given token $z_i$ from other high-density tokens. We identify those tokens with relatively high $\rho_i \times \delta_i$ as cluster centers and then allocate other tokens to their nearest cluster center according to the Euclidean distances. Finally, we utilize the average token within each cluster to represent the corresponding cluster. The vision region of the merged token is the union of the vision regions within the corresponding cluster.

**Temporal Visual Token Merging.** To adapt the dynamic visual tokens to video, we extend the visual tokens across frames. However, directly consolidating all frames into a limited number of visual tokens may lead to the loss of temporal information within the video. For example, in Fig. 2, the video demonstrates the process of cooking pasta before preparing the sauce. Simply merging all frames would pose challenges for the model in determining the correct sequence, such as whether to prepare the sauce first, cook the pasta first, or simultaneously cook the pasta while preparing the sauce. Therefore, we propose temporal visual token merging to first divide the video into several critical events. After that, we make the visual tokens only expand over frames within the same event.

Given the $m_{th}$ frame $\mathbb{Z}^m = \{z_i^m\}_{i=1}^{L}$ of a video, we first apply mean-pooling over all tokens to obtain the frame-level representation $f^m$. Similar to the spatial visual token merging method, we leverage DPC-KNN to amalgamate non-essential frames.

Specifically, we first compute the local density $\rho^m$ and the distance index $\delta^m$ of each frame $f^m$. Then, we identify those frames with relatively high $\rho^m \times \delta^m$ as cluster centers and then allocate other frames to their nearest cluster center according to the Euclidean distances. We treat each cluster as a critical event and denote the set of indexes of the frames in the cluster as $\mathbb{F}$. Therefore, the set of visual tokens within the $n_{th}$ event $\mathbb{F}_n$ can be formulated as:

$$\tilde{\mathbb{Z}}_n = \big\{z_i^m | m \in \mathbb{F}_n,\ i \in \{1, 2, ..., L\}\big\}. \tag{3}$$

After completing the temporal visual token merging, we obtain the set of visual tokens within the event, i.e., $\tilde{\mathbb{Z}}$. To make the visual tokens expand over frames within the event, we adjust Eq. 1 and Eq. 2 in the spatial visual token merging method to the following form:

$$\tilde{\rho}_i = \exp\Big(-\frac{1}{K}\sum_{z_k \in \text{KNN}(z_i,\tilde{\mathbb{Z}})} \|z_k - z_i\|^2\Big), \quad \tilde{\delta}_i = \begin{cases} \min_{j:\tilde{\rho}_j > \tilde{\rho}_i} \|z_j - z_i\|^2, & \text{if } \exists j \text{ s.t. } \tilde{\rho}_j > \tilde{\rho}_i. \\ \max_j \|z_j - z_i\|^2, & \text{otherwise.} \end{cases} \tag{4}$$

Finally, we concatenate the expanded visual tokens together in order of events to ensure the broader temporal understanding required for videos.

**Multi-scale Representation.** To further enhance the capabilities of our model, we propose a multi-step aggregation method designed to provide multi-scale visual features for LLMs. Specifically, in Chat-UniVi, the initial visual tokens at the first merging step are derived from the vision encoder

Table 1: **GPT-based evaluation for image understanding.** Following Liu et al. (2023), we report the relative scores to GPT-4 for instruction-following questions. "†" denotes our own re-implementation of LLaVA under our training settings (excluding video data) for a fair comparison.

| Methods | LLM Size | Visual Tokens | Conversation | Detail Description | Complex Reasoning | All |
|---|---|---|---|---|---|---|
| LLaVA | 13B | 256 | 83.1 | 75.3 | 96.5 | 85.1 |
| LLaVA | 7B | 256 | 70.3 | 56.6 | 83.3 | 70.1 |
| LLaVA† | 7B | 256 | 78.8 | 70.2 | 91.8 | 80.4 |
| Chat-UniVi | 7B | **112** | **84.1** | **74.2** | **93.7** | **84.2** |

Table 2: **GPT-based evaluation for video understanding.** Following Maaz et al. (2023), we report the relative scores between the output of the model and the ground truth, with the assistance of GPT. It is worth noting that the results reported in Maaz et al. (2023) span a range from 0 to 5. To standardize the metrics, we normalize all scores to a scale of 0 to 100.

| Methods | LLM Size | Correctness of Information | Detail Orientation | Contextual Understanding | Temporal Understanding | Consistency |
|---|---|---|---|---|---|---|
| Video-LLaMA | 7B | 39.2 | 43.6 | 43.2 | 36.4 | 35.8 |
| LLaMA-Adapter | 7B | 40.6 | 46.4 | 46.0 | 39.6 | 43.0 |
| VideoChat | 7B | 44.6 | 50.0 | 50.6 | 38.8 | 44.8 |
| Video-ChatGPT | 7B | 48.0 | 50.4 | 52.4 | 39.6 | 47.4 |
| Chat-UniVi | 7B | **57.8** | **58.2** | **69.2** | **57.8** | **56.2** |

of CLIP. Then, we progressively merge visual tokens with similar semantic meanings and obtain different numbers of tokens in different steps. The higher-level features encompass abstract semantic concepts, while the lower levels emphasize representations of visual details. In practice, we execute a three-step aggregation process for each input image or video. Finally, we concatenate the outputs from each merging step and utilize a trainable projection matrix $W$ to transform these multi-scale visual features into language embedding tokens, which serve as inputs for LLMs.

It is worth noting that despite this concatenation, the number of visual tokens in our method remains significantly lower than the original visual tokens initially generated by the vision transformer.

### 3.2 MULTIMODAL TRAINING SCHEME

**Multimodal Pre-training.** Following previous works (Liu et al., 2023), our training is divided into two stages. In the first stage, we pre-train the projection matrix $W$ while freezing both the LLM and the vision encoder. This strategic freezing of the LLM empowers our method to effectively capture semantic visual information without any discernible compromise in the performance of LLMs.

**Joint Instruction Tuning.** After completing the first stage, the model is able to understand human queries but still fails to generate reasonable and coherent linguistic responses. In the second stage, we fully fine-tune the large language model and the projection matrix $W$ on a multimodal instruction-following dataset. This dataset is a composite of multi-turn conversations and single-turn conversations presented in a conversational format, alongside single images, multiple images, and videos as visual input. Through joint training on the mixture dataset, Chat-UniVi achieves a superior comprehension of a wide array of directives and produces more natural and dependable output. More encouragingly, Chat-UniVi possesses the unique capability to directly handle both images and videos without necessitating any realignment between the vision and language models.

## 4 EXPERIMENTS

### 4.1 EXPERIMENTAL SETUP

**Model Settings.** Following previous works (Liu et al., 2023), we adopt the vision encoder of CLIP (ViT-L/14) (Radford et al., 2021) as the visual foundation model. We chose an instruction-tuned variant of LLaMA2 (Touvron et al., 2023b), i.e., Vicuna (Team, 2023), as our language foundation model. Specifically, we utilize the Vicuna-v1.5 model, comprised of 7B parameters.

**Data and Training Details.** For the multimodal pre-training stage, we utilize the image-caption pairs from various datasets, including COCO (Chen et al., 2015) and CC3M-595K screened from CC3M (Sharma et al., 2018) by LLaVA (Liu et al., 2023). We pre-train Chat-UniVi for one epoch

Table 3: **Zero-shot and fine-tuning question answering accuracy on the ScienceQA test set.** Question classes: NAT = natural science, SOC = social science, LAN = language science, TXT = text context, IMG = image context, NO = no context, G1-6 = grades 1-6, G7-12 = grades 7-12. "†" denotes our own re-implementation of LLaVA under our training settings (excluding video data).

| Methods | LLM Size | Subject | | | Context Modality | | | Grade | | Average |
|---|---|---|---|---|---|---|---|---|---|---|
| | | NAT | SOC | LAN | TXT | IMG | NO | G1-6 | G7-12 | |
| Random Choice | - | 40.28 | 46.13 | 29.25 | 47.45 | 40.08 | 33.66 | 39.35 | 40.67 | 39.83 |
| Human | - | 90.23 | 84.97 | 87.48 | 89.60 | 87.50 | 88.10 | 91.59 | 82.42 | 88.40 |
| *Zero-shot Question Answering Accuracy (%)* | | | | | | | | | | |
| GPT-4 | 1T+ | 84.06 | 73.45 | 87.36 | 81.87 | 70.75 | 90.73 | 84.69 | 79.10 | 82.69 |
| GPT-3 | 175B | 74.64 | 69.74 | 76.00 | 74.44 | 67.28 | 77.42 | 76.80 | 68.89 | 73.97 |
| LLaVA† | 7B | 47.78 | 41.96 | 53.64 | 47.90 | 44.03 | 51.92 | 49.63 | 45.29 | 48.08 |
| Chat-UniVi | 7B | **58.61** | **61.08** | **61.82** | **57.33** | **58.25** | **61.39** | **62.04** | **56.23** | **59.96** |
| *Fine-tuning Question Answering Accuracy (%)* | | | | | | | | | | |
| LLaVA | 13B | 90.36 | 95.95 | 88.00 | 89.49 | 88.00 | 90.66 | 90.93 | 90.90 | 90.92 |
| LLaVA† | 7B | 79.71 | 91.68 | 82.82 | 80.94 | 83.24 | 81.46 | 83.74 | 81.74 | 83.02 |
| LLaMA-Adapter | 7B | 84.37 | 88.30 | 84.36 | 83.72 | 80.32 | 86.90 | 85.83 | 84.05 | 85.19 |
| LLaMA-SciTune | 7B | 84.50 | **94.15** | 82.91 | 88.35 | 83.64 | **88.74** | 85.05 | 85.60 | 86.11 |
| Chat-UniVi | 7B | **88.50** | 93.03 | **85.91** | **88.51** | **85.97** | 88.15 | **88.88** | **88.60** | **88.78** |

Table 4: **Zero-shot video question answering accuracy.** We follow the evaluation protocol in Maaz et al. (2023), i.e., employing GPT-assisted evaluation to assess the capabilities of models. "Score" denotes the confidence score from 0 to 5 assigned by the GPT model.

| Methods | LLM Size | MSRVTT-QA | | MSVD-QA | | TGIF-QA | | ActivityNet-QA | |
|---|---|---|---|---|---|---|---|---|---|
| | | Accuracy | Score | Accuracy | Score | Accuracy | Score | Accuracy | Score |
| FrozenBiLM | 1B | 16.8 | - | 32.2 | - | 41.0 | - | 24.7 | - |
| Video-LLaMA | 7B | 29.6 | 1.8 | 51.6 | 2.5 | - | - | 12.4 | 1.1 |
| LLaMA-Adapter | 7B | 43.8 | 2.7 | 54.9 | 3.1 | - | - | 34.2 | 2.7 |
| VideoChat | 7B | 45.0 | 2.5 | 56.3 | 2.8 | 34.4 | 2.3 | 26.5 | 2.2 |
| Video-ChatGPT | 7B | 49.3 | 2.8 | 64.9 | 3.3 | 51.4 | 3.0 | 35.2 | 2.7 |
| Chat-UniVi | 7B | **54.6** | **3.1** | **65.0** | **3.6** | **60.3** | **3.4** | **45.8** | **3.2** |

with a batch size of 128, employing the AdamW (Kingma & Ba, 2014; Loshchilov & Hutter, 2017) optimizer with a cosine schedule. The learning rate is set to 2e-3, and the warm-up rate is 0.03. For the joint instruction tuning stage, we incorporate multimodal instruction data from multiple sources: (i) multimodal in-context instruction datasets, such as MIMIC-IT (Li et al., 2023a; Antol et al., 2015; Hudson & Manning, 2019), (ii) visual instruction datasets, such as LLaVA, (iii) video instruction data from Video-ChatGPT (Maaz et al., 2023). All input images or frames are resized to $224 \times 224$. We train Chat-UniVi for 2 epochs with a batch size of 128, and the learning rate is set to 2e-5.

## 4.2 GPT-BASED EVALUATION

**Image Understanding.** To quantitatively measure the image understanding capability, we report the GPT-4 evaluation results in Tab. 1. Following Liu et al. (2023); Zhang et al. (2023c), we employ 90 questions based on 30 COCO validation images, covering various aspects, including conversation, detail description, and complex reasoning. We utilize the GPT-4 model to evaluate the outputs of the model in these three aspects, as well as provide an overall score. For a comprehensive description of image understanding metrics, please refer to the appendix. As shown in Tab. 1, Chat-UniVi uses fewer visual tokens while achieving superior performance. Notably, our method, even as a 7B model, can achieve the performance level of a 13B model, demonstrating the effectiveness of our method.

**Video Understanding.** To quantitatively measure the video understanding capability, we report the GPT evaluation results in Tab. 2. Following Maaz et al. (2023), we employ a test set based on the ActivityNet dataset (Caba Heilbron et al., 2015) and utilize the GPT-3.5 model to assign a relative score to the outputs of the model in the following five aspects: Correctness of Information, Detail Orientation, Contextual Understanding, Temporal Understanding, and Consistency. Please refer to the appendix for more details. As shown in Tab. 2, Chat-UniVi, even as a unified model, significantly surpasses recently proposed state-of-the-art methods, such as VideoChat and Video-ChatGPT, that exclusively focus on video, which demonstrates the effectiveness of our method.

Table 5: **Zero-shot object hallucination evaluation on the COCO validation set.** "Yes" represents the proportion of positive answers that the model outputs. "†" denotes our own re-implementation of LLaVA under our training settings (excluding video data) for a fair comparison.

| Methods | LLM Size | Random | | | Popular | | | Adversarial | | |
|---|---|---|---|---|---|---|---|---|---|---|
| | | Accuracy | F1-Score | Yes | Accuracy | F1-Score | Yes | Accuracy | F1-Score | Yes |
| LLaVA | 13B | 64.12 | 73.38 | 83.26 | 63.90 | 72.63 | 81.93 | 58.91 | 69.95 | 86.76 |
| MiniGPT-4 | 13B | 79.67 | 80.17 | 52.53 | 69.73 | 73.02 | 62.20 | 65.17 | 70.42 | 67.77 |
| InstructBLIP | 13B | 88.57 | 89.27 | 56.57 | 82.77 | 84.66 | 62.37 | 72.10 | 77.32 | 73.03 |
| MM-GPT | 7B | 50.10 | 66.71 | 99.90 | 50.00 | 66.67 | 100.00 | 50.00 | 66.67 | 100.00 |
| mPLUG-Owl | 7B | 53.97 | 68.39 | 95.63 | 50.90 | 66.94 | 98.57 | 50.67 | 66.82 | 98.67 |
| LLaVA† | 7B | 72.16 | 78.22 | 76.29 | 61.37 | 71.52 | 85.63 | 58.67 | 70.12 | 88.33 |
| Chat-UniVi | 7B | **85.19** | **86.05** | **54.67** | **69.50** | **74.39** | **69.10** | **64.97** | **71.54** | **73.10** |

Table 6: **Ablation study about the multi-scale representation.** "Detail" denotes the "Detail Description" in the context of image understanding or "Detail Orientation" in the context of video understanding. For image understanding, "Reason" denotes the "Complex Reasoning". For video understanding, "Correct", "Context", and "Temporal" stand for "Correctness of Information", "Contextual Understanding", and "Temporal Understanding", respectively.

| Methods | Image Understanding | | | | Video Understanding | | | | |
|---|---|---|---|---|---|---|---|---|---|
| | Conversation | Detail | Reason | All | Correct | Detail | Context | Temporal | Consistency |
| Single-scale | 70.5 | 63.4 | 88.3 | 74.2 | 54.6 | 56.4 | 65.8 | 52.8 | 52.2 |
| Multi-scale | **84.1** | **74.2** | **93.7** | **84.2** | **57.8** | **58.2** | **69.2** | **57.8** | **56.2** |

Table 7: **Ablation study about instruction tuning scheme.** "Only Image" indicates training solely on image data. "Image + Video" means training on image data followed by fine-tuning on video data. "Image & Video" denotes training on a combined dataset of both image and video data.

| Methods | Image Understanding | | | | Video Understanding | | | | |
|---|---|---|---|---|---|---|---|---|---|
| | Conversation | Detail | Reason | All | Correct | Detail | Context | Temporal | Consistency |
| Only Image | 84.0 | 69.3 | 89.3 | 81.5 | 43.4 | 48.6 | 56.8 | 45.4 | 46.2 |
| Only Video | 72.7 | 55.8 | 71.5 | 66.8 | 57.4 | **58.8** | 69.0 | 56.4 | 56.0 |
| Image + Video | 45.5 | 31.3 | 76.1 | 50.9 | 51.2 | 55.6 | 64.8 | 50.0 | 50.4 |
| Video + Image | 79.0 | 69.2 | 88.5 | 79.1 | 45.6 | 49.8 | 58.2 | 46.4 | 47.8 |
| Image & Video | **84.1** | **74.2** | **93.7** | **84.2** | **57.8** | 58.2 | **69.2** | **57.8** | **56.2** |

## 4.3 QUESTION-ANSWER EVALUATION

**ScienceQA Performance.** ScienceQA (Lu et al., 2022) is a comprehensive multimodal science question-answering dataset comprising 21k multiple-choice questions. Each example in ScienceQA contains a visual context, a textual context, a question, multiple options, and the correct answer. For the input of Chat-UniVi, we concatenate the question, textual context, and options sequentially into a single sentence. We report both zero-shot and fine-tuning results in Tab. 3. As shown in Tab. 3, Chat-UniVi shows competitive performance across all metrics. Notably, Chat-UniVi outperforms LLaMA-SciTune (Horawalavithana et al., 2023), a model specifically tailored for science question answering, which fully demonstrates the superiority of our method.

**Zero-shot Video-question Answering Performance.** In Tab. 4, we show the zero-shot video-question answering performance on several commonly used open-ended question-answer datasets, including MSRVTT-QA (Xu et al., 2017), MSVD-QA (Xu et al., 2017), TGIF-QA FrameQA (Jang et al., 2017), and ActivityNet-QA (Yu et al., 2019). Our evaluation protocol follows that of Maaz et al. (2023), utilizing GPT-assisted evaluation to assess the capabilities of models. As shown in Tab. 4, Chat-UniVi outperforms the recently proposed state-of-the-art methods, e.g., FrozenBiLM (Yang et al., 2022) and Video-ChatGPT, across various datasets. Chat-UniVi exhibits a slight improvement on MSVD-QA. We attribute this to the short duration of videos in MSVD-QA, which may not fully showcase the advantages of our method in temporal modeling.

## 4.4 OBJECT HALLUCINATION EVALUATION

In Tab. 5, we report the results of the polling-based object probing evaluation (Li et al., 2023d). For details of the polling-based object probing evaluation, please refer to the appendix. As shown in Tab. 5,

Table 8: **Ablation study about the number of spatial visual clusters.** "$C_1$", "$C_2$", and "$C_3$" denote the number of clusters at the first step, the second step, and the last step, respectively.

| $C_1$ | $C_2$ | $C_3$ | Visual Tokens | Conversation | Detail description | Complex reasoning | All |
|---|---|---|---|---|---|---|---|
| 16 | 8 | 4 | 28 | 78.6 | 69.0 | **95.1** | 81.1 |
| 32 | 16 | 8 | 56 | 82.7 | 67.2 | 94.5 | 81.6 |
| 64 | 32 | 16 | 112 | **84.1** | **74.2** | 93.7 | **84.2** |
| 128 | 64 | 32 | 224 | 79.8 | 68.7 | 83.8 | 79.8 |

Table 9: **Ablation study about the number of temporal visual clusters.** "$M$" is the frame length. "$1/M$" denotes that the model directly consolidates all frames into a single event.

| Temporal Clustering Ratio | Correctness of Information | Detail Orientation | Contextual Understanding | Temporal Understanding | Consistency |
|---|---|---|---|---|---|
| $1/M$ | 51.2 | 41.8 | 47.6 | 32.8 | 42.2 |
| $1/32$ | 57.2 | 58.0 | **69.6** | 56.2 | 54.2 |
| $1/16$ | **57.8** | **58.2** | 69.2 | **57.8** | 56.2 |
| $1/8$ | 56.8 | **58.2** | 68.0 | 55.8 | **57.8** |

(a) Comparison of image-based conversations

(b) Comparison of video-based conversations

Figure 3: **Human evaluations on multimodal conversations.** In 30 image conversation scenarios and 30 video conversation scenarios, the evaluators rate the model on a scale of 0 to 10 based on its multimodal conversation performance. Finally, we use the average score as the final model score.

Chat-UniVi outperforms the recently proposed state-of-the-art methods, such as MultiModal-GPT (MM-GPT). Notably, as a 7B model, our method even outperforms the 13B model, e.g., MiniGPT-4, in the object hallucination evaluation. We attribute this success to the multi-scale representation that equips our method to perceive both high-level semantic concepts and low-level visual appearance.

## 4.5 ABLATIVE ANALYSIS

**Effect of the Multi-scale Representation.** To investigate the impact of the multi-scale representation of our method, we provide the ablation results in Tab. 6. Multi-scale representation improves both image understanding and video understanding of the model. These results provide evidence for the benefits of employing a multi-scale representation in multimodal large language models.

**Effect of the Tuning Scheme.** In Tab. 7, we provide the ablation study on the instruction tuning scheme. We find that visual instruction tuning using only one type of medium, such as images, results in a decrease in comprehension of another medium, such as videos. However, pre-training on one medium and fine-tuning on another may lead to knowledge degradation from the pre-training stage. In contrast, our joint training strategy, which involves training on a mixed dataset of images and videos, endows the model with the capability to process both types of visual inputs. Among all tuning schemes, joint training consistently achieves the highest performance, confirming its effectiveness.

**Effect of the Number of Spatial Visual Clusters.** To explore the influence of the number of spatial visual clusters, we provide the ablation results in Tab. 8. We find that a smaller number of visual clusters may decrease the capacity to grasp fine visual details, whereas a larger number of visual clusters may introduce redundancy and potentially reduce the overall performance of the model. To strike a balance between detailed understanding and model learning complexity, we set the number of clusters at the three levels to 64, 32, and 16 respectively in practice.

**Effect of the Number of Temporal Visual Clusters.** Videos vary in length, with longer videos typically containing more events. Therefore, in Chat-UniVi, the number of temporal visual clusters is determined proportionally based on the number of input video frames. As shown in Tab. 9, we find that a smaller clustering ratio may result in the loss of crucial temporal information within the video. Conversely, a larger clustering ratio increases the computational overhead of the model. We observe

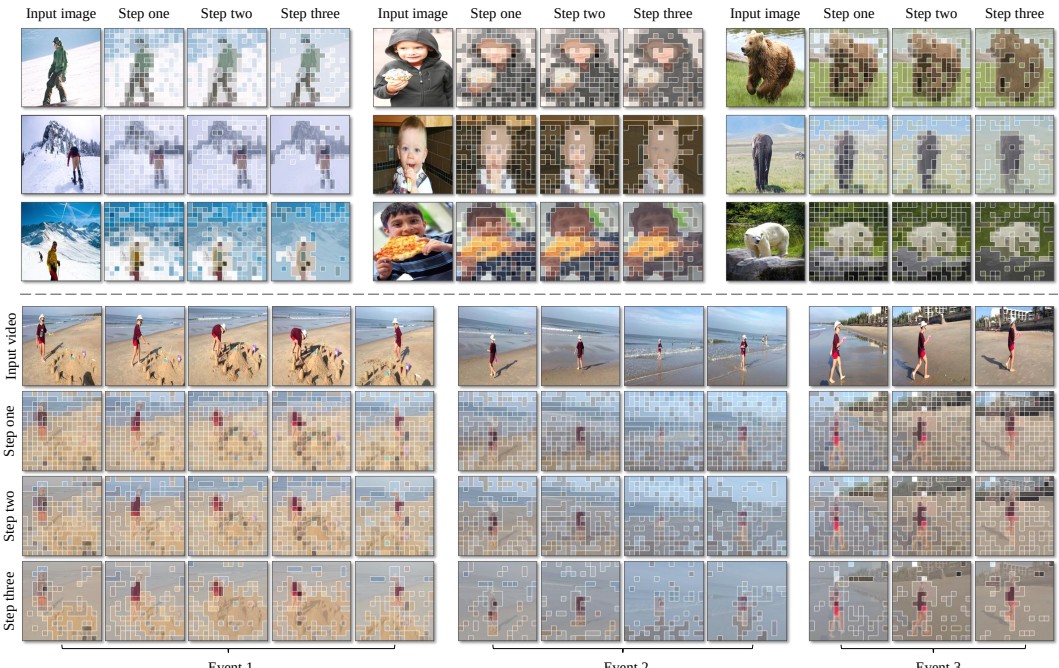

Figure 4: **Visualization of the dynamic visual tokens.** More visualizations of the dynamic visual tokens are shown in Fig. A and Fig. B. Examples of conversations are provided in Appendix. E.

that the model performs optimally when the clustering ratio is set to $1/16$. Therefore, in practice, we adopt a default temporal clustering ratio of $1/16$ for better performance.

### 4.6 QUALITATIVE ANALYSIS

**Human Evaluation.** In our evaluation, we manually assess the performance of Chat-UniVi and baselines in 30 image conversation scenarios and 30 video conversation scenarios. The results are presented in Fig. 3. OpenFlamingo (Awadalla et al., 2023), derived from Flamingo (Alayrac et al., 2022), and Otter (Li et al., 2023a), an in-context instruction tuning variant of OpenFlamingo, are also included in our comparison. As shown in Fig. 3, we find that methods based on Flamingo exhibit limitations in their ability to comprehend videos. This limitation is attributed to their use of a query transformer to extract a fixed number of visual tokens from videos of varying lengths, which hinders their effectiveness in modeling temporal comprehension. In contrast, Chat-UniVi, functioning as a unified model, not only outperforms methods built upon the Flamingo but also surpasses models specifically designed for image (e.g., LLaVA) and video (e.g., Video-ChatGPT).

**Visualization of the Dynamic Visual Tokens.** We provide the visualization in Fig. 4 and invite readers to explore more visualizations in the appendix. It is important to emphasize that our proposed token merging method is parameter-free and operates without the need for object outline labels. As shown in Fig. 4, the proposed dynamic visual tokens effectively generalize objects and backgrounds. This capability enables Chat-UniVi to reconcile the intricate spatial nuances of images with the broader temporal understanding required for videos with a limited number of visual tokens.

## 5 CONCLUSION

In this paper, we introduce Chat-UniVi, a unified multimodal large language model designed to proficiently comprehend and engage in conversations about both images and videos. To seamlessly bridge the intricate spatial nuances of images with the broader temporal understanding required for videos, we propose a unified representation framework employing dynamic visual tokens. This novel representation leverages DPC-KNN to progressively cluster visual tokens and provides multi-scale features. More encouragingly, Chat-UniVi is trained on a mixed dataset encompassing both images and videos, enabling it to be directly applicable to tasks involving both media types without necessitating any modifications. Extensive experimental results demonstrate that Chat-UniVi, as a unified model, consistently surpasses even methods exclusively designed for images or videos.

REPRODUCIBILITY STATEMENT

1. For data details.
   (a) We outline the composition of the training data in Section 4.1.
   (b) We describe in detail the composition of the training data, as well as our data filtering method in Appendix B.
   (c) We provide a detailed description of the training data in Tab. C.
   (d) We promise to release a data download link upon publication, which can directly download the data we have processed.

2. For model settings.
   (a) We outline the model settings in Section 4.1.
   (b) We describe in detail the model settings in Appendix B.
   (c) We also experiment with other model settings, such as another vision encoder. The results are provided in Tab. E.

3. For training hyperparameters.
   (a) We outline the training hyperparameters in Section 4.1.
   (b) We describe in detail the training hyperparameters in Appendix B.
   (c) We also provide detailed training hyperparameters for fine-tuning our model on the ScienceQA dataset in Appendix B.

4. For code.
   (a) We have attached the code to the supplementary material.
   (b) In this code, we also provide the pre-trained model weights and the process of the evaluation of the proposed method.
   (c) Besides, we provide the additional demo code, providing an interactive interface to make it easier for readers to experience the capabilities of our model.
   (d) We promise to release a more detailed and clean code version upon publication.

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

# A APPENDIX

**Abstract**   This appendix provides additional discussions (Appendix A), implementation details (Appendix B), several additional experiments (Appendix C), additional visualization results (Appendix D), more qualitative analysis (Appendix E), and details of quantitative evaluations (Appendix F).

**Code**   We have attached the code to the supplementary material. In this code, we also provide the pre-trained model weights and the process of the evaluation of the proposed method. We promise to release a more detailed and clean code version upon publication.

**Demo**   In the supplementary material, we provide the additional demo code, providing an interactive interface to make it easier for readers to experience the capabilities of our model.

# A ADDITIONAL DISCUSSIONS

## A.1 COMPARISON OF CHAT-UNIVI AND OTHER METHODS

Existing methods often focus exclusively on either image or video inputs. Recently, there have also been some methods (Alayrac et al., 2022; Wu et al., 2023b; Chen et al., 2023) that support both images and videos, and they can be broadly divided into two classes.

- **Q-former based methods.** The first class of methods uses a query transformer to extract a fixed number of tokens for each image and video. These methods are exemplified by Flamingo (Alayrac et al., 2022), OpenFlamingo (Awadalla et al., 2023), and Otter (Li et al., 2023a). However, videos vary in length, posing a challenge for these methods, as they extract a fixed number of visual tokens from each video, limiting their ability to effectively capture temporal comprehension. Human evaluation results (see Fig. 3) also substantiate that these methods struggle to strike a balance between image and video comprehension.

- **Multi-encoder methods.** The second category of methods employs separate pre-trained image and video encoders to process images and videos independently. Prominent examples of this approach include X-LLM (Chen et al., 2023) and NExT-GPT (Wu et al., 2023b). However, these methods introduce redundancy within the model and present difficulties when trained jointly. Most importantly, this approach does not leverage the advantages of joint training with both image and video data. Consequently, they do not align with our primary objective of developing a unified vision-language model.

In contrast to the previous works, Chat-UniVi uniformly represents images and videos using multi-scale dynamic visual tokens. The proposed Chat-UniVi has two compelling advantages:

- **Variable length video features.** In Chat-UniVi, the number of temporal visual clusters is determined proportionally based on the number of input video frames. In contrast to the Q-former based methods, Chat-UniVi allocates a greater number of visual tokens to longer videos. Therefore, our method is better suited for variable-length video understanding.

- **Unified visual encoder.** Chat-UniVi employs a shared visual encoder to consistently process both images and videos. In contrast to multi-encoder methods, our method eliminates the need for introducing redundant parameters and streamlines the training process.

- **Benefit from joint training.** Due to the unified representation framework for both images and videos, Chat-UniVi can be trained on mixed datasets that include both images and videos. This allows for direct application to tasks involving both images and videos. Most importantly, we find that this joint training strategy can simultaneously enhance the model's understanding of both images and videos. Experimental results are shown in Tab. 7.

In Tab. A, we show the comparison of Chat-UniVi and other methods. For Q-former based methods, the advantages of joint training are not shown, and even the performance of the model may affect each other when multiple datasets are mixed (Alayrac et al., 2022). However, the potential to benefit from joint training cannot be ruled out. In addition, the multi-encoder method can also select a video encoder that can encode dynamic length features.

Table A: **Comparison with other methods.** "✗" denotes that the model does not have this property. "✔" denotes that the model has this property. "–" indicates a temporary lack of experimental evidence.

| Type | Methods | Variable Length Features | Unified Visual Encoder | Benefit from Joint Training |
|---|---|---|---|---|
| Q-former based methods | Flamingo OpenFlamingo, Otter | ✗ | ✔ | – |
| Multi-encoder methods | X-LLM, NExT-GPT | – | ✗ | ✗ |
| Unified methods | Chat-UniVi | ✔ | ✔ | ✔ |

Table B: **Comparison of Chat-UniVi and another token clustering method.** "✗" denotes that the model does not have this property. "✔" denotes that the model has this property.

| Methods | Parameter-free | Video Input | Image Understanding | | | |
|---|---|---|---|---|---|---|
| | | | Conversation | Detail | Reason | All |
| Ma et al. (2023) | ✗ | ✗ | 71.8 | 60.9 | 91.6 | 75.0 |
| Chat-UniVi | ✔ | ✔ | **84.1** | **74.2** | **93.7** | **84.2** |

## A.2 COMPARISON OF CHAT-UNIVI AND OTHER CLUSTERING TRANSFORMER METHODS

There have also been recent methods (Ma et al., 2023; Xu et al., 2022; Zeng et al., 2022; Jin et al., 2023a) to explore the role of token clustering within the transformer framework. However, none of these methods can be directly extended to video, and additional parameters need to be trained. We summarize the advantages of our method as follows:

- **Supporting video input.** In contrast to other methods, Chat-UniVi extends the tokens clustering method to incorporate video inputs, achieving the integration of image and video representations for the first time. Our work is the first to demonstrate that this unified representation can reconcile the intricate spatial details of images with the broader temporal understanding required for videos.

- **Without parameters.** Our clustering method is parameter-free and therefore requires no training. Interestingly, we find that this parameter-free clustering method serves as the linchpin to the success of our model. As shown in Tab. B, the performance of the clustering method with training parameters is significantly inferior to the parameter-free clustering method we propose. We attribute this phenomenon to the gradient instability in multimodal conversation training, which hinders the convergence of parameterized methods.

## A.3 LIMITATIONS AND FUTURE WORK

In this section, we delineate the limitations of our work and outline avenues for future research.

**The Enduring Impact of Large Language Models.** Our method leverages the strength of pre-trained Large Language Models, and as a consequence, also inherits their vulnerabilities.

- **Hallucination.** While our experiments (see Tab. 5) demonstrate the effectiveness of our method in addressing hallucinations, it is important to acknowledge that the issue of hallucinations in LLMs remains a challenge yet to be fully resolved. The phenomenon of illusory responses in LLMs can result in unsupported conjectures during open multimodal conversations, and addressing this issue has the potential to significantly expedite advancements in the field. For a more in-depth exploration of common weaknesses observed in large LLMs, please refer to Brown et al. (2020); Rae et al. (2021).

- **Long sequence processing.** Transformer-based language models often exhibit suboptimal generalization when confronted with test sequences considerably longer than their training data (Press et al., 2022). This becomes particularly evident in multi-turn conversations, where the model may exhibit forgetfulness of prior conversational context, resulting in erroneous

Table C: **Description of training data.** "✘" denotes that the dataset does not have this property. "✔" denotes that the dataset has this property. "‡" represents the dataset filtered from MIMIC-IT, containing exclusively image data. In order to further filter the training data, we also delete the duplicate data in LLaVA-instruct-150K and MIMIC-IT.

| Datasets | Image Inputs | Video Inputs | Multi-turn Conversations | Number of Conversations |
|---|---|---|---|---|
| *Multimodal Pre-training Stage* | | | | |
| CC3M-595K | ✔ | ✘ | ✘ | 595K |
| COCO | ✔ | ✘ | ✘ | 956K |
| *Joint Instruction Tuning Stage* | | | | |
| LLaVA-instruct-150K | ✔ | ✘ | ✔ | 150K |
| MIMIC-IT-399K‡ | ✔ | ✘ | ✘ | 399K |
| Video-ChatGPT-instruct | ✘ | ✔ | ✘ | 100K |

responses. Simultaneously, we find a decline in model performance when multiple videos are inputted, which could also be attributed to constraints associated with sequence length.

- **Prompt sensitivity.** In-context learning has demonstrated disconcerting sensitivity to various aspects of demonstrations, including prompt formats (Zhao et al., 2021). Notably, different prompt formats can yield entirely contradictory output results. Finding a solution to this issue holds the potential to greatly accelerate progress in the field.

**Natural Language Output.** Natural language serves as a robust and adaptable input/output interface for describing visual tasks to the model, facilitating the generation of outputs, or estimating conditional probabilities for potential outcomes. However, it may prove to be a less convenient interface for tasks that require conditioning on or predicting more structured outputs, such as bounding boxes, as well as for generating dense pixel predictions. Besides, the flexibility of the natural language output also makes it difficult to evaluate the performance of the model.

**More Modalities.** Future work can explore alternative modalities, such as audio, in addition to visual inputs. The incorporation of multiple modalities holds the promise of broadening the spectrum of tasks that the model can address, and it has the potential to enhance their performance by leveraging synergies among these various modalities. For example, contemplating audio information alongside video processing can significantly augment the video understanding of the model.

# B    IMPLEMENTATION DETAILS

**Data Details.** For the multimodal pre-training stage, we utilize the image-caption pairs from various datasets, including COCO (Chen et al., 2015) and CC3M-595K screened from CC3M (Sharma et al., 2018) by LLaVA (Liu et al., 2023). All input images are resized to $224 \times 224$. For the joint instruction tuning stage, we incorporate multimodal instruction data from multiple sources: (i) multimodal in-context instruction datasets, such as MIMIC-IT (Li et al., 2023a; Antol et al., 2015; Hudson & Manning, 2019), (ii) visual instruction datasets, such as LLaVA, (iii) video instruction data from Video-ChatGPT (Maaz et al., 2023). In order to further filter the training data, we delete the duplicate data in LLaVA-instruct-150K and MIMIC-IT, and delete the video data in MIMIC-IT. This dataset is a composite of multi-turn conversations and single-turn conversations presented in a conversational format, alongside single images, multiple images, and videos as visual input. For each video, we select 64 frames as input for the model. All input images or frames are resized to $224 \times 224$. We provide a detailed description of the training data in Tab. C.

**Model Settings.** Following previous works (Liu et al., 2023), we adopt the vision encoder of CLIP (ViT-L/14) (Radford et al., 2021) as the visual foundation model. We chose an instruction-tuned variant of LLaMA2 (Touvron et al., 2023b), i.e., Vicuna (Team, 2023), as our language foundation model. Specifically, we utilize the Vicuna-v1.5 model, comprised of 7B parameters.

**Training Hyperparameters.** For the multimodal pre-training stage, we pre-train Chat-UniVi for one epoch with a batch size of 128, employing the AdamW optimizer with a cosine schedule. The learning rate is set to 2e-3, and the warm-up rate is 0.03. For the joint instruction tuning stage, we

Table D: **Comparison between the LoRA and full fine-tuning.** "Detail" denotes the "Detail Description" in the context of image understanding or "Detail Orientation" in the context of video understanding. For image understanding, "Reason" denotes the "Complex Reasoning". For video understanding, "Correct", "Context", and "Temporal" stand for "Correctness of Information", "Contextual Understanding", and "Temporal Understanding", respectively.

| Methods | Image Understanding | | | | Video Understanding | | | | |
|---|---|---|---|---|---|---|---|---|---|
| | Conversation | Detail | Reason | All | Correct | Detail | Context | Temporal | Consistency . |
| LoRA | 76.1 | 68.6 | 82.4 | 75.8 | 52.8 | 55.0 | 63.8 | 51.6 | 53.8 |
| Full fine-tuning | **84.1** | **74.2** | **93.7** | **84.2** | **57.8** | **58.2** | **69.2** | **57.8** | **56.2** |

Table E: **Comparison between the EVA CLIP and the Openai CLIP.** We choose EVA-CLIP (ViT-G), which has a similar number of parameters as Openai-CLIP (ViT-L/14), for the experiment.

| Methods | Image Understanding | | | | Video Understanding | | | | |
|---|---|---|---|---|---|---|---|---|---|
| | Conversation | Detail | Reason | All | Correct | Detail | Context | Temporal | Consistency |
| EVA-CLIP | 80.0 | 74.7 | 91.2 | 82.1 | 57.2 | 58.8 | 67.8 | 55.2 | 54.6 |
| Openai-CLIP | **84.1** | **74.2** | **93.7** | **84.2** | **57.8** | **58.2** | **69.2** | **57.8** | **56.2** |

Table F: **Effect of the multi-scale representation on object hallucination.** "Yes" represents the proportion of positive answers that the model outputs.

| POPE | Methods | LLM Size | Accuracy | Precision | Recall | F1-Score | Yes |
|---|---|---|---|---|---|---|---|
| Random | Single-scale | 7B | 73.88 | 67.03 | **97.06** | 79.30 | 74.63 |
| | Multi-scale | 7B | **85.19** | **83.59** | 88.66 | **86.05** | **54.67** |
| Popular | Single-scale | 7B | 56.36 | 53.50 | **97.20** | 69.01 | 90.83 |
| | Multi-scale | 7B | **69.50** | **64.10** | 88.60 | **74.39** | **69.10** |
| Adversarial | Single-scale | 7B | 55.63 | 53.07 | **97.26** | 68.67 | 91.63 |
| | Multi-scale | 7B | **64.97** | **60.23** | 88.06 | **71.54** | **73.10** |

train Chat-UniVi for 2 epochs with a batch size of 128, and the learning rate is set to 2e-5, employing the AdamW optimizer with a cosine schedule. The warm-up rate is set to 0.03.

**ScienceQA Fine-tuning Settings.** We start with a pre-trained model to fine-tune. We fine-tune the model for 9 epochs with a batch size of 32, employing the AdamW optimizer with a cosine schedule. The learning rate is set to 2e-5, and the warm-up rate is 0.03.

# C    ADDITIONAL EXPERIMENTS

**Comparison between the LoRA and Full Fine-tuning.** When the number of model parameters is too large, full fine-tuning of retraining all model parameters becomes expensive, so many recent methods freeze most of the model parameters and train the model with LoRA (Hu et al., 2022). We provide the results of the comparison between the LoRA and full fine-tuning in Tab. D. We find that LoRA can achieve competitive performance with full fine-tuning while saving more than half the GPU memory required for training. Future work can use LoRA to extend our method on larger LLMs and vision encoders to achieve better performance.

**Analysis of the Vision Encoder.** EVA-CLIP (Sun et al., 2023) is a recently developed multimodal model with performance comparable to Openai-CLIP (Radford et al., 2021). We provide the results of the comparison between EVA-CLIP and Openai-CLIP in Tab. E. We find that the performance of EVA-CLIP is comparable to that of Openai-CLIP when the number of parameters is equal. However, EVA-CLIP offers a larger version of the model with a parameter count of 1.8B, so we think it might be better to adopt a larger EVA-CLIP than Openai-CLIP when using larger LLMs.

Table G: **Detailed results on object hallucination evaluation.** "†" denotes our own re-implementation of LLaVA under our training settings (excluding video data) for a fair comparison.

| POPE | Methods | LLM Size | Accuracy | Precision | Recall | F1-Score | Yes |
|------|---------|----------|----------|-----------|--------|----------|-----|
| Random | LLaVA | 13B | 64.12 | 59.38 | 95.99 | 73.38 | 83.26 |
| | MiniGPT-4 | 13B | 79.67 | 78.24 | 82.20 | 80.17 | 52.53 |
| | InstructBLIP | 13B | 88.57 | 84.09 | 95.13 | 89.27 | 56.57 |
| | MultiModal-GPT | 7B | 50.10 | 50.05 | **100.00** | 66.71 | 99.90 |
| | mPLUG-Owl | 7B | 53.97 | 52.07 | 99.60 | 68.39 | 95.63 |
| | LLaVA† | 7B | 72.16 | 78.22 | 76.29 | 78.22 | 76.29 |
| | Chat-UniVi | 7B | **85.19** | **83.59** | 88.66 | **86.05** | **54.67** |
| Popular | LLaVA | 13B | 63.90 | 58.46 | 95.86 | 72.63 | 81.93 |
| | MiniGPT-4 | 13B | 69.73 | 65.86 | 81.93 | 73.02 | 62.20 |
| | InstructBLIP | 13B | 82.77 | 76.27 | 95.13 | 84.66 | 62.37 |
| | MultiModal-GPT | 7B | 50.00 | 50.00 | **100.00** | 66.67 | 100.00 |
| | mPLUG-Owl | 7B | 50.90 | 50.46 | 99.40 | 66.94 | 98.57 |
| | LLaVA† | 7B | 61.37 | 56.63 | 97.00 | 71.52 | 85.63 |
| | Chat-UniVi | 7B | **69.50** | **64.10** | 88.60 | **74.39** | **69.10** |
| Adversarial | LLaVA | 13B | 58.91 | 55.11 | 95.72 | 69.95 | 86.76 |
| | MiniGPT-4 | 13B | 65.17 | 61.19 | 82.93 | 70.42 | 67.77 |
| | InstructBLIP | 13B | 72.10 | 65.13 | 95.13 | 77.32 | 73.03 |
| | MultiModal-GPT | 7B | 50.00 | 50.00 | **100.00** | 66.67 | 100.00 |
| | mPLUG-Owl | 7B | 50.67 | 50.34 | 99.33 | 66.82 | 98.67 |
| | LLaVA† | 7B | 58.67 | 54.90 | 97.00 | 70.12 | 88.33 |
| | Chat-UniVi | 7B | **64.97** | **60.23** | 88.06 | **71.54** | **73.10** |

**Effect of the Multi-scale Representation on Object Hallucination.** As shown in Tab. 5, Chat-UniVi, as a 7B model, even outperforms the 13B model, e.g., MiniGPT-4, in the object hallucination evaluation. We attribute this success to the multi-scale representation that equips our method to perceive both high-level semantic concepts and low-level visual appearance. In Tab. F, we show the results of ablation experiments on object hallucination evaluation for the multi-scale representation. We find that multi-scale representation improves the ability to resist hallucinations. Therefore, multi-scale representation is beneficial for multimodal LLMs.

**Detailed Results on Object Hallucination Evaluation.** In Tab. G, we report the detailed results of the polling-based object probing evaluation (Li et al., 2023d). As shown in Tab. G, Chat-UniVi outperforms the recently proposed state-of-the-art methods. Notably, as a 7B model, our method even outperforms the 13B model, e.g., MiniGPT-4, in the object hallucination evaluation. These results demonstrate the effectiveness of our method.

# D  ADDITIONAL VISUALIZATION RESULTS

**Visualization of the dynamic visual tokens for the image inputs.** To gain a deeper insight into the functionality of our proposed dynamic visual tokens, we present the additional visualization results for the image inputs in Fig. A. In Fig. A, we provide a diverse range of visualizations encompassing various image categories, including portraits, sports, wildlife, art, architecture, and food. It is crucial to underscore that our proposed token merging method operates without the need for object outline labels and is parameter-free. As shown in Fig. A, the proposed dynamic visual tokens effectively generalize objects and backgrounds, empowering Chat-UniVi to capture the spatial nuances of images using a limited number of visual tokens.

**Visualization of the dynamic visual tokens for the video inputs.** To gain a more comprehensive understanding of our proposed dynamic visual tokens, we also present additional visualization results for the video inputs in Fig. B. In the case of videos, the video is initially divided into several events, and subsequently, these visual tokens expand over frames within each event to encapsulate frame-level dynamics. Notably, our method imposes no restrictions on the number of frames per event, showcasing the remarkable flexibility and generalization ability of our methodology. As shown in Fig. B, the proposed dynamic visual tokens significantly reduce the number of visual tokens while

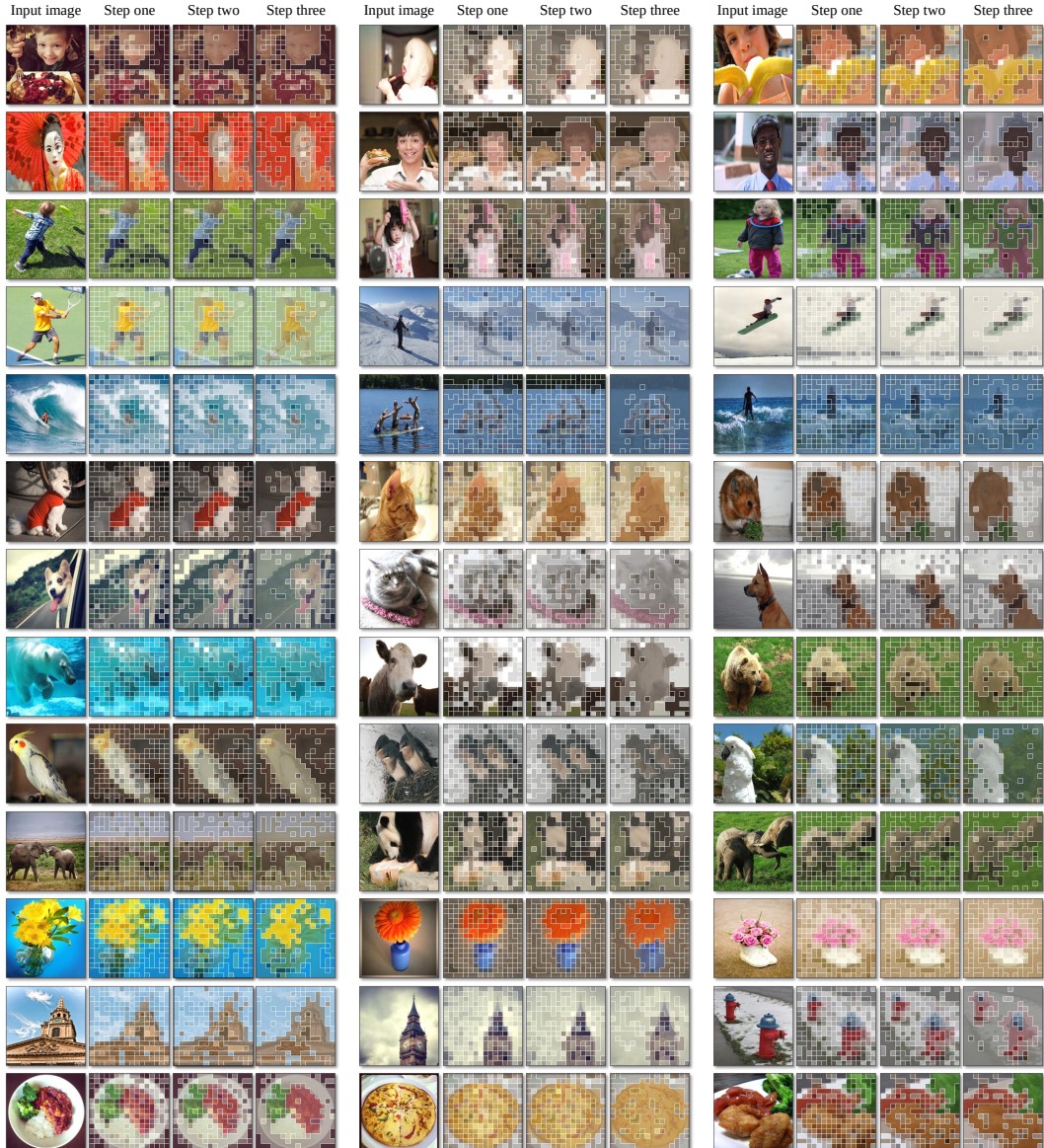

Figure A: **Visualization of the dynamic visual tokens for the image inputs.** We provide a diverse range of visualizations encompassing various image categories, including portraits, sports, wildlife, art, architecture, and food. It is important to emphasize that our proposed token merging method is parameter-free and operates without the need for object outline labels.

maintaining the expressive capabilities of the model. This empowerment equips Chat-UniVi with the capacity to capture the broader temporal understanding required for videos, all within the confines of a limited number of visual tokens.

# E  ADDITIONAL QUALITATIVE ANALYSIS

**The conversation includes both the image and the video.**    In Fig. C and Fig. D, we present examples of conversations that encompass both the image and the video. As shown in Fig. C and Fig. D, Chat-UniVi offers detailed and contextually appropriate responses aligned with user prompts. These illustrative examples showcase the remarkable ability of Chat-UniVi to comprehend both image and video contexts across multiple conversational turns.

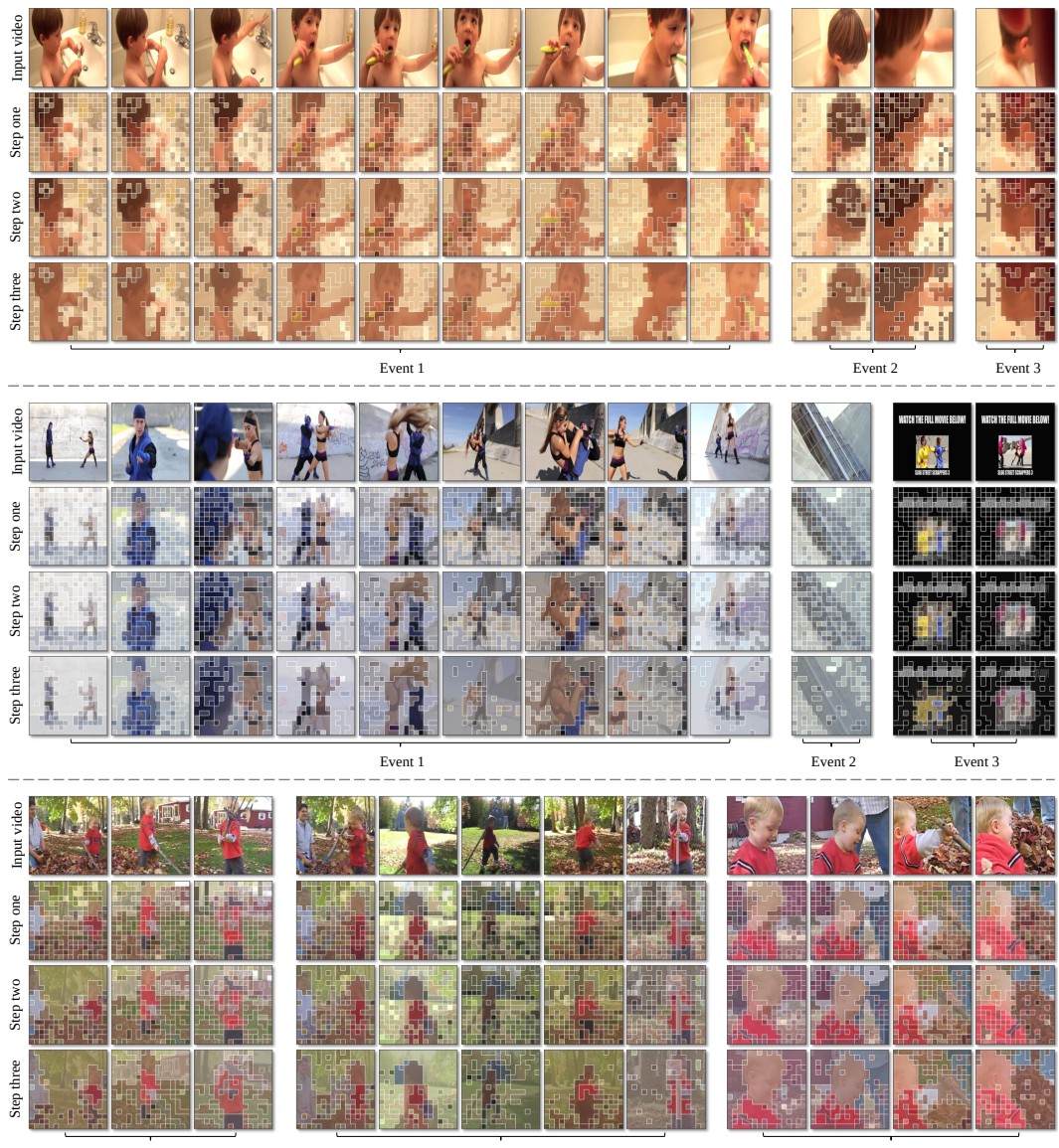

Figure B: **Visualization of the dynamic visual tokens for the video inputs.** It is important to emphasize that our proposed token merging method is parameter-free and operates without the need for object outline labels. Our method imposes no restrictions on the number of frames per event, showcasing the remarkable flexibility and generalization ability of our methodology.

**The conversation includes multiple videos.** Fig. E illustrates a conversation example including multiple videos. As shown in Fig. E, Chat-UniVi can use the information of multiple videos in the context, and provide appropriate and coherent responses based on user prompts. The illustrative example showcases the remarkable ability of Chat-UniVi to comprehend multiple video contexts across multiple conversational turns.

**The conversation includes multiple images.** Fig. F provides an illustrative conversation example including multiple images. As shown in Fig. F, Chat-UniVi adeptly leverages information from multiple images within the context, enabling it to make choices among various images. This illustrative example highlights the impressive capacity of Chat-UniVi to grasp multiple image contexts seamlessly throughout various conversational exchanges.

**The conversation includes the image.** Fig. G features an example of a conversation that incorporates an image. As shown in Fig. G, Chat-UniVi excels at providing detailed descriptions and

can even craft compelling narratives inspired by the image. The illustrative example showcases the remarkable ability of Chat-UniVi in the realms of reasoning and creative expression.

**The conversation includes the video.** In Fig. H and Fig. I, we offer examples of conversations that incorporate the video. As shown in Fig. H and Fig. I, Chat-UniVi exhibits a remarkable proficiency in comprehending videos and is adept at offering valuable insights inspired by the video content. These illustrative examples showcase the remarkable ability of Chat-UniVi to grasp video contexts and engage in reasoned responses.

# F DETAILS OF QUANTITATIVE EVALUATIONS

**GPT-based Evaluation For Image Understanding.** Our quantitative evaluation protocol follows that of Liu et al. (2023). Following Liu et al. (2023); Zhang et al. (2023c), we employ 90 questions based on 30 COCO validation images, covering various aspects, including conversation, detail description, and complex reasoning. These images are randomly selected by Liu et al. (2023). We utilize the GPT-4 model to generate reference responses based on the question, and the ground-truth bounding boxes and captions. During the model evaluation process, the model predicts answers based on both the question and input image. After obtaining the response from the model, we feed the question, visual information (in the format of captions and bounding boxes), the generated response, and the reference response to GPT-4. GPT-4 evaluates the helpfulness, relevance, accuracy, and level of detail of the responses, assigning an overall score on a scale of 1 to 10, where a higher score indicates better overall performance. Besides, we also ask GPT-4 to provide a comprehensive explanation of the evaluation to enhance our understanding of the models.

**GPT-based Evaluation For Video Understanding.** The quantitative evaluation protocol for video understanding follows the methodology introduced by Maaz et al. (2023). Specifically, Maaz et al. (2023) curates a test set based on the ActivityNet-200 dataset (Caba Heilbron et al., 2015), which includes videos with rich, dense descriptive captions and associated question-answer pairs from human annotations. During the model evaluation process, we employ the GPT-3.5 model to assign a relative score to the generated predictions on a scale of 1-5, across five critical aspects: (1) Correctness of information. (2) Detail orientation. (3) Contextual understanding. (4) Temporal understanding. (5) Consistency. It is worth noting that the results reported in Maaz et al. (2023) span a range from 0 to 5. To standardize the metrics, we normalize all scores to a scale of 0 to 100.

**Zero-shot Video Question Evaluation.** Our evaluation protocol follows that of Maaz et al. (2023), utilizing GPT-assisted evaluation to assess the capabilities of models. During the model evaluation process, we feed the question, the ground-truth answer, and the generated response to the GPT-3.5 model. GPT-3.5 evaluates whether the generated responses are correct and assigns a matching score on a scale of 0 to 5, where a higher score indicates better overall performance.

**Zero-shot Object Hallucination Evaluation.** To quantitatively evaluate the hallucination problem of the model, we adopt the polling-based object probing evaluation (POPE) process proposed by Li et al. (2023d). Specifically, POPE formulates the evaluation of object hallucination as a binary classification task, where the model is prompted to respond with either "Yes" or "No" to queries like "Is there a chair in the image?". Li et al. (2023d) randomly selects 500 images from the COCO validation set. Each image contains more than three ground-truth objects in the annotations, and six questions are generated for each image. The annotations of objects in images directly construct the questions with the answer "Yes". For the questions with the answer "No", three different strategies are employed for sampling their probing objects as follows:

- **Random Sampling.** Randomly sampling objects that do not exist in the image.
- **Popular Sampling.** Selecting the top-3 most frequently occurring objects in the COCO dataset that are absent from the image.
- **Adversarial Sampling.** Initially, Li et al. (2023d) rank all objects based on their co-occurring frequencies with the ground-truth objects, and subsequently select the top-3 most frequent objects from this list that are not present in the image.

Tell me about the movie.

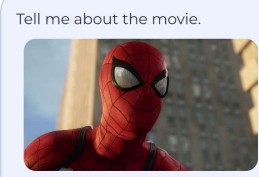

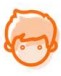

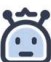

The movie is a computer-animated superhero film featuring the iconic character Spiderman. It is set in a bustling city where Spiderman, a superhero with incredible spider-like abilities, fights crime and protects the innocent. The film showcases Spiderman's adventures as he navigates the urban landscape, using his spider-sense and web-shooting abilities to take down villains and save lives. The movie captures the essence of the Spiderman character, combining action, adventure, and humor, while also exploring themes of responsibility and heroism.

Write a rich narration for this video based on the background of the movie.

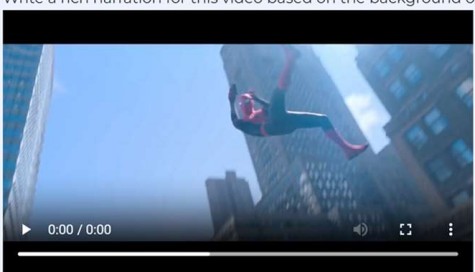

0:00 / 0:00

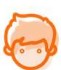

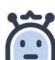

In a bustling metropolis, a man dressed in a red and black Spiderman suit, clutching a backpack, swings through the air, defying gravity. He soars above the busy city streets, where a multitude of vehicles, including cars, trucks, and buses, navigate through the chaos. The Spiderman suit stands out against the urban backdrop, a symbol of hope and heroism in a world of darkness. The city's towering buildings and neon signs create a dazzling backdrop for Spiderman's daring feats, as he leaps from rooftop to rooftop, taking on the city's criminal underworld. The city's inhabitants look up in awe and wonder at the spectacle unfolding before their eyes, as the Spiderman legend continues to inspire and captivate generations to come.

If Spiderman and Batman were to make a movie together, what do you think the story would be?

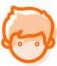

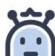

If Spiderman and Batman were to make a movie together, it would be a thrilling and action-packed adventure that would showcase their unique abilities and teamwork. The story could revolve around a common enemy, such as a powerful villain who poses a threat to both the city of New York and Gotham City. Spiderman and Batman would have to work together to uncover the villain's plans and put an end to their evil schemes.

The story could begin with Spiderman swinging through the streets of New York, trying to track down a lead on the villain. Meanwhile, in Gotham City, Batman is investigating a series of strange occurrences that seem to be connected to the same villain. As they both close in on the villain, they discover that they are working together and must put aside their differences to stop them.

Throughout the movie, we could see the dynamic between Spiderman and Batman, highlighting their unique approaches to fighting crime and their contrasting personalities. Spiderman, being a web-slinging hero, would rely on his agility and quick reflexes, while Batman would use his detective skills and martial arts prowess to take down the villain.

The climax of the movie would involve a thrilling showdown between Spiderman and Batman and the villain, with both heroes putting their lives on the line to save the city. In the end, they would emerge victorious, having not only saved the day but also forged a newfound respect and admiration for each other's abilities. The movie would conclude with a final shot of the two heroes, standing together, ready to protect the city from any threat that may come their way.

Figure C: **An example of a conversation that includes both the image and the video.** The blue box shows the user input. The gray box shows the model output.

Can you write a short poem inspired by this video?

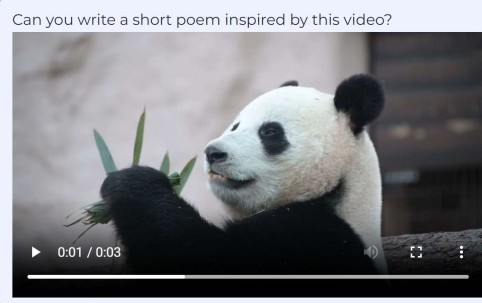

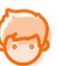

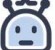

Sure, here's a short poem inspired by the video:

A panda bear in a zoo,
Eating bamboo with a smile anew.
It reaches up with a paw,
To grab some greens to munch on now.

The zookeeper feeds it bamboo,
And the panda bear enjoys it so.
It chews and grinds the plant,
And savors every single crumb.

A panda bear in a zoo,
A simple life, yet so true.
It eats its greens with glee,
And finds joy in the small things, you see.

If I want to use this image for the zoo's publicity, do you have any suggestions for designing the poster?

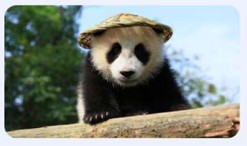

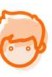

Yes, I have a few suggestions for designing a poster using this image. Since the image features a panda bear wearing a hat, you could use this unique and eye-catching visual to create a memorable and playful poster. Here are a few ideas:

1. Keep it simple: Use a clean and minimalist design to highlight the panda bear's adorable appearance and the hat. You could use a large photo of the panda bear with the hat in the center of the poster, and keep the text to a minimum.
2. Add a tagline: Include a short and catchy tagline, such as "Experience the cuteness overload" or "Visit the zoo and meet the cutest animal," to capture the viewer's attention and encourage them to visit the zoo.
3. Use bold colors: Choose bold and vibrant colors to make the poster stand out, such as using the panda bear's natural colors as a base and adding contrasting colors for the hat and background.
4. Add more visuals: To make the poster more dynamic, you could include additional visuals, such as other animals wearing hats or interacting with the panda bear, fun facts about pandas, or images of zoo visitors enjoying their time with the panda bear.
5. Include a call-to-action: Encourage viewers to visit the zoo by including a clear call-to-action, such as "Visit us today to see this adorable panda bear in person" or "Share this poster with your friends and family to spread the cuteness."

By incorporating these suggestions, you can create an eye-catching and memorable poster that showcases the panda bear's unique appearance and encourages zoo visitors to experience the cuteness for themselves.

Figure D: **An example of a conversation that includes both the image and the video.** The blue box shows the user input. The gray box shows the model output.

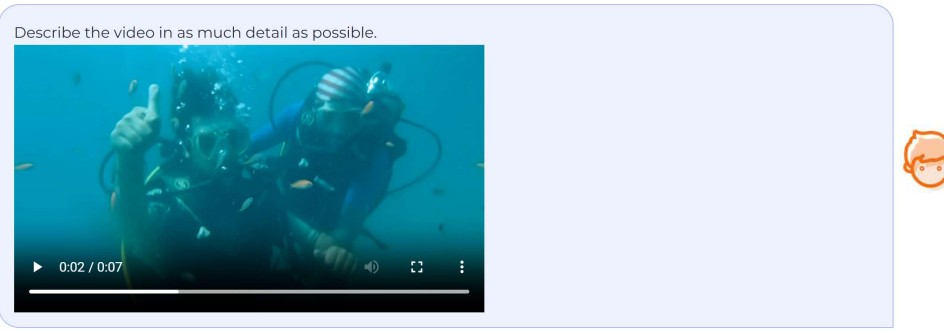

Describe the video in as much detail as possible.

0:02 / 0:07

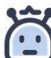

The video shows two scuba divers underwater, exploring the beautiful and diverse ecosystem of the ocean. They are seen swimming around, observing the marine life and admiring the coral reefs. The scuba divers are wearing proper diving gear, including oxygen tanks, masks, and wetsuits. The water is crystal clear, allowing the divers to see the vibrant colors of the fish and other marine creatures. The coral reefs are home to a variety of species, including colorful fish, sea turtles, and sea anemones. The divers move slowly and deliberately, ensuring they do not disturb the delicate ecosystem. The video captures the peaceful and serene atmosphere of the underwater world, with the sound of the divers' bubbles and the gentle lapping of the waves providing a soothing background sound. Overall, the video offers a mesmerizing glimpse into the beauty and complexity of the ocean's ecosystem, and the importance of preserving it for future generations.

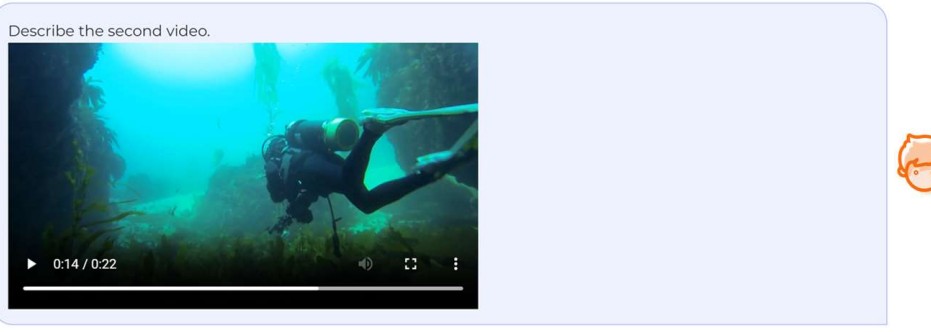

Describe the second video.

0:14 / 0:22

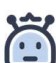

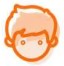

The second video shows a group of scuba divers exploring the ocean floor. They are seen swimming around and observing the marine life. The video captures the divers' movements as they move around the ocean floor, and the camera follows them closely, providing a clear view of the underwater environment. The divers are equipped with scuba gear, and they appear to be in awe of the vibrant and diverse marine life surrounding them. The video showcases the beauty of the underwater world and the importance of preserving it.

If I want to splice these two videos together, do you have any suggestions?

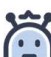

Yes, I can provide some suggestions for splicing these two videos together. First, you can start with the first video and show the scuba divers swimming around the coral reef and observing the marine life. Then, you can transition to the second video and show the divers swimming around the ocean floor and observing the diverse marine life. You can use a smooth transition effect to blend the two videos seamlessly. After that, you can transition back to the first video and show the divers swimming around the coral reef and observing the marine life. Finally, you can end the video with a final shot of the divers swimming away from the camera, leaving the viewers with a sense of awe and appreciation for the beauty of the underwater world.

Figure E: **An example of a conversation that includes multiple videos.** The blue box shows the user input. The gray box shows the model output.

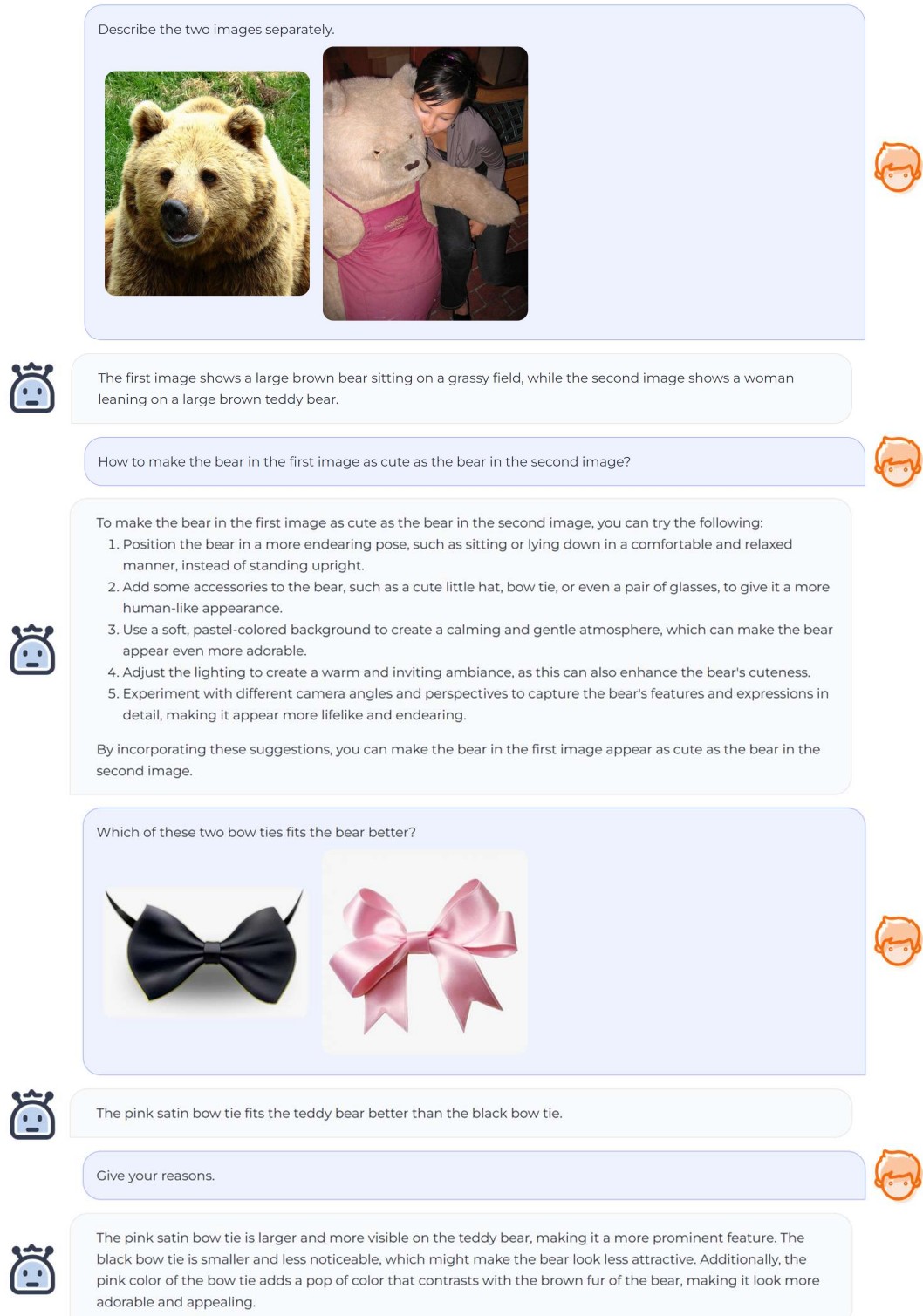

Figure F: **An example of a conversation that includes multiple images.** The blue box shows the user input. The gray box shows the model output.

How many superheroes are there in this image?

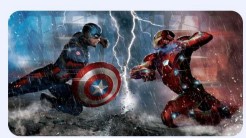

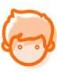

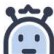

There are two superheroes in this image.

Describe each of them in detail.

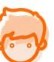

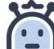

In the image, there are two superheroes engaged in an action-packed fight. One of them is wearing a red and yellow suit, which is a classic representation of the iconic superhero Iron Man. Iron Man is known for his high-tech armor and powerful weapons, as well as his intelligence and wit. He is a founding member of the Avengers and has been portrayed in various media, including movies and comic books.

The other superhero is wearing a blue and red costume, which is reminiscent of the character Captain America. Captain America is a symbol of patriotism and is known for his exceptional strength, endurance, and moral values. He is also a founding member of the Avengers and has been portrayed as a key figure in various Marvel Comics storylines.

Both superheroes are depicted in mid-air, leaping towards each other with great force, creating an intense and exciting scene. Their clash highlights their incredible abilities and showcases their epic battle in the comic book or movie from which this image is taken.

Write a movie story based on this image.

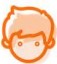

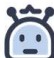

The movie opens with a dramatic scene of the two superheroes, Iron Man and Captain America, leaping towards each other in mid-air. The camera captures their intense expressions and the determination in their eyes as they engage in a fierce battle. The audience is left wondering about the reason behind their fight and what could be at stake.

As the story unfolds, we learn that the conflict between Iron Man and Captain America stems from a disagreement on how to handle a dangerous threat that has emerged. Iron Man believes that the use of force is necessary to protect humanity, while Captain America holds a more idealistic view and seeks to resolve the situation through peaceful means.

Their battle takes them across various locations, from the streets of a bustling city to the depths of a secret underground facility. Along the way, they encounter other superheroes who are caught in the crossfire and must choose sides. The tension and drama continue to build as the story progresses, with each character revealing their motivations and backstory.

As the climax approaches, the two superheroes come face to face in an epic showdown. They engage in a thrilling fight, showcasing their unique abilities and skills. In the end, they realize that their conflict has brought them closer to the truth and that their differences have made them stronger. They put aside their differences and unite to face the common enemy, saving the world and restoring peace.

The movie concludes with a final scene that highlights the importance of teamwork and the power of unity. The superheroes stand together, ready to face any challenge that comes their way, while the audience is left with a sense of awe and admiration for these extraordinary characters.

Figure G: **An example of a conversation that includes the image.** The blue box shows the user input. The gray box shows the model output.

What is the video about?

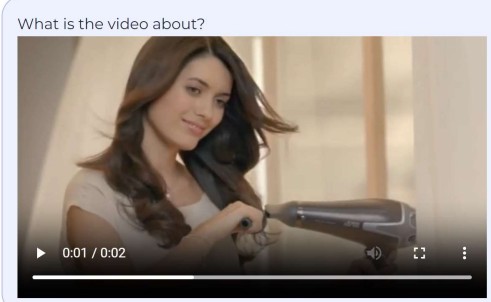

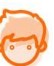

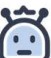

The video is about a woman who is blow drying her hair in a room.

Generate a creative advertising feature about the hair dryer used by the girl, which can be used on the product website.

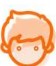

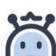

The hair dryer used by the girl in the video is not just any ordinary hair dryer, it's a smart hair dryer that comes with a built-in intelligence. This smart hair dryer uses advanced technology to analyze the hair type and drying conditions, ensuring that the hair is dried evenly and efficiently. The smart hair dryer also comes with a built-in temperature and airflow control system, allowing the user to customize the drying experience to their hair type.

With the help of the smart hair dryer, the girl in the video was able to achieve a sleek and polished look in just a few minutes. The hair dryer is also lightweight and easy to use, making it a great option for busy women on the go.

In addition to its advanced features, the smart hair dryer is also environmentally friendly. It uses less energy and produces less heat than traditional hair dryers, making it a more sustainable option.

To learn more about this amazing hair dryer and its features, visit the product website. With its advanced technology and eco-friendly design, this smart hair dryer is the perfect tool for any woman looking to achieve a sleek and polished look.

If this video is a commercial for this hair dryer, do you have any suggestions to make this video more popular?

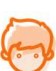

To make this video more popular, we can consider the following suggestions:
1. Include before-and-after shots: Showing the difference in the girl's hair before and after using the hair dryer can help viewers understand the effectiveness of the product.
2. Add testimonials: Including testimonials from satisfied customers who have used the hair dryer can help build trust and credibility for the product.
3. Highlight unique features: Emphasize the unique features of the hair dryer, such as the smart technology, temperature control, and airflow adjustment, to showcase its advantages over other hair dryers.
4. Showcase different hair types and styles: Demonstrate how the hair dryer works on different hair types and styles, showcasing its versatility and adaptability.
5. Include a call-to-action: Encourage viewers to visit the product website or make a purchase by including a clear call-to-action at the end of the video.
6. Optimize for social media: Create short clips or teasers from the video to share on social media platforms, driving traffic to the product website and increasing brand awareness.

By incorporating these suggestions, the video can become more engaging, informative, and persuasive, ultimately leading to increased interest and sales of the hair dryer.

Figure H: **An example of a conversation that includes the video.** The blue box shows the user input. The gray box shows the model output.

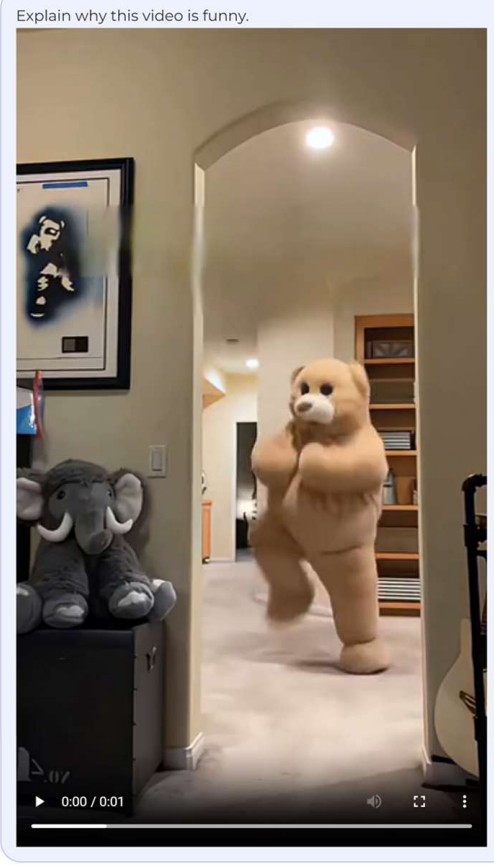
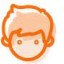

Explain why this video is funny.

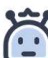

The video is funny because it features a person wearing a large, humorous teddy bear costume. The individual's choice to wear this oversized and comical outfit creates a sense of amusement and playfulness, which is amplified when they begin to dance. The combination of the exaggerated size of the costume and the person's decision to dance adds a lighthearted and entertaining element to the video. This unconventional and whimsical scene captures the viewer's attention and evokes a sense of joy and laughter, making it an enjoyable and humorous experience.

Do you have any suggestions to make this video more popular?

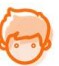

To make this video more popular, several strategies could be employed. Here are a few suggestions:

1. Social Media Sharing: Encourage the person in the teddy bear costume to share the video on their social media accounts, tagging friends and family to help expand its reach. Utilizing hashtags relevant to the content can also increase visibility to a broader audience.
2. Collaborations: Collaborate with other content creators, influencers, or popular personalities to feature the video or create a crossover content. This can help attract their followers and expose the video to a larger audience.
3. Paid Advertising: Invest in targeted online ads or promotions, such as Google AdWords or social media ads, to drive traffic to the video. This can be an effective way to reach a specific demographic or location, increasing the video's visibility and potential viewership.
4. Optimize Video Content: Ensure that the video is well-edited, visually appealing, and includes engaging elements such as captions, special effects, or additional footage. This can help capture the viewer's attention and keep them engaged throughout the video.
5. Engage with the Audience: Encourage viewers to like, comment, and share the video. Responding to comments and interacting with the audience can help create a sense of community and increase the video's popularity by organically spreading the content through word-of-mouth.

By implementing these strategies, the video's popularity can be increased, potentially leading to more views, shares, and engagement with the content.

Figure I: **An example of a conversation that includes the video.** The blue box shows the user input. The gray box shows the model output.

