# OpenReview forum: "Chat-UniVi: A Unified Vision-Language Model for Image and Video Understanding"
_ICLR.cc/2024/Conference — ICLR 2024 Conference Withdrawn Submission_

### Official Review · Reviewer_aENx · 2023-10-21

**Soundness:** 2 fair
**Presentation:** 3 good
**Contribution:** 2 fair
**Rating:** 5
**Confidence:** 4

**Summary:**

The paper proposes Chat-Univi, a unified VLM to handle both images and videos. To seamlessly bridge the intricate spatial nuances of images with the broader temporal understanding required for videos, the paper proposes a dynamic visual token design and takes multi-scale spatial and temporal information into account. Experimental results show that the performance of the designed model is promising.

**Strengths:**

1. While LLM has been widely explored, the VLM is still relatively unexploited. The paper is timely focusing on an important topic with high value.

2. Integrating image and video modality together to form the VLM is how the VLM should be eventually designed in my point of view. I'm glad to see that the paper is proposing the idea along this direction.

3. Experimental results are overall promising.

3. The paper is clearly presented and well-written.

**Weaknesses:**

1. The paper may lack novelty/scientific contribution to hit the ICLR acceptance bar. The joint image / video learning is important but not quite innovative. The idea of multi-scale dynamic visual tokens is simple. While a simple solution is good, if its effectiveness and generalizability are solidly proved. With certain missing results and analysis (mentioned below), I don't feel the effectiveness and generalizability of the design is soundly proved in the current version.

2. Some important results/discussions are missing:
(a) The comparison to image based VLM is not thorough. In Table 1 and Table 3- zero-shot part, the paper only compares Chat-UniVi with LLaVA, GPT-3/GPT-4, but not other SOTA models such as mPLUG-Owl, Flamingo, etc. A more thorough comparison to SOTA models should be added into the paper. Also, while the paper follows the LLAVA setting to evaluate the model on only 90 questions in Table 1, I still feel the setting is too small to achieve any solid conclusion on model generalizability. A more thorough evaluation on larger benchmarks would help provide more convincing results.

(b) It's unclear that the improvement of the performance is from joint image/video learning or from the multi-scale dynamic visual tokens. If from both, what percentage of gain does each of them contribute? There should be an ablation on this.

(c) There needs to be an ablation on number of steps. Why 3-steps is a good setup?

(d) The explanation on why Chat-Univi performs better on hallucination (in Table 5) is ambiguous ("We attribute this success to the multi-scale representation that equips our method to perceive both high-level semantic concepts and low-level visual appearance"). Why perceiving both high-level and low-level information can help reduce hallucination? If multi-scale presentation is helpful in hallucination, there should be an ablation on it (hallucination performance with / without multi-scale), which is missing now.

**Questions:**

1. One thing is unclear to me: in each critical event, are all the tokens clustered regardless of the frames? From Eq (4), it seems that all the tokens are clustered regardless of the frames (e.g., tokens from different frames can be clustered together), but from Figure 2, it seems that tokens are only clustered within each frame but not cross the frames (in step 1, 2, 3, the concept "frame" still exist). Please help clarify.

2. Why not using object detection / segmentation methods for the spatial visual token merging, instead of using the KNN? Using detection / segmentation may help provide more accurate and semantic meaningful clusters.

3. When aggregating the tokens within each cluster, why not using weighted average instead of simply average? Will weighted average provide more robust / representative tokens?

**Details Of Ethics Concerns:**

LLM models are often with certain ethics issues. The paper is currently missing a discussion on discrimination / bias / fairness concerns of the proposed model.

---

### Official Review · Reviewer_ThZs · 2023-10-31

**Soundness:** 3 good
**Presentation:** 3 good
**Contribution:** 3 good
**Rating:** 5
**Confidence:** 4

**Summary:**

This paper presents a unified image and video model geared towards chat applications. Visual tokens from a frozen CLIP-backed vision
encoder are dynamically merged using clustering at multiple-scales, and are then passed to a large language model after a simple projection layer. The projection layers are first fine-tuned on captioning datasets and later instruction-tuned along with the LLM on multimodal image and video datasets. Several experiments relating to image and video understanding, zero-shot question answering, object hallucination etc. along with some human evaluation results are presented and the proposed method outperforms similar sized video and image specific models.

**Strengths:**

This paper is well-written and easy to follow. This paper demonstrates that a simple parameter-free clustering method can effectively sparsify the image and video features and such features can serve as strong inputs to LLM. Furthermore, the paper shows that they can jointly instruction-tune the LLM on both image and video datasets. Experimental results demonstrate superior performance against competing image and video based methods. The clustering visualizations show a clear semantic structure and the visualized chat applications in the appendix show a very good grounding.

**Weaknesses:**

1. The paper makes a strong claim that it is the first successful unified vision-language model that can outperform image and video models. However, there is broad work on such vision-language models [1, 2, 3, 4] that can do well on both image and video tasks.
2. Only GPT based evaluations are considered making it difficult to situate with SOTA.
- Image and video captioning results are not presented. See [1]
- Standard image and video classification results such as on ImageNet and Kinetics are not presented. See [1]
- GPT-based score for QA tasks such as MSRVTT, ActivityNet are presented making it difficult to compare with other methods.
3. The approach of sparsifying transformer inputs is not novel [2, 5, 6, 7]. Note that [5] also uses clustering.

Refs:
[1]: Flamingo: a Visual Language Model for Few-Shot Learning
[2]: MaMMUT: A Simple Architecture for Joint Learning for MultiModal Tasks
[3]: Alternating Gradient Descent and Mixture-of-Experts for Integrated Multimodal Perception
[4]: OmniVL: One Foundation Model for Image-Language and Video-Language Tasks
[5]: Not All Tokens Are Equal: Human-centric Visual Analysis via Token Clustering Transformer
[6]: Rethinking Video ViTs: Sparse Video Tubes for Joint Image and Video Learning
[7]: DynamicViT: Efficient Vision Transformers with Dynamic Token Sparsification

**Questions:**

Is the token sparsification necessary for efficiency purpose or for performance?
How do the QA results compare with Flamingo?
Is there a way to do captioning and compare ROUGE, CiDER scores?
How does the model do on ImageNet and Kinetics classification?

---

### Official Review · Reviewer_iewy · 2023-11-01

**Soundness:** 3 good
**Presentation:** 3 good
**Contribution:** 3 good
**Rating:** 6
**Confidence:** 5

**Summary:**

This paper presents a new video-based model using instruction-tuning. Their key idea is to design a new architecture that leverages mult-scale features and dynamic visual tokens. The authors first conduct a pre-training using COCO and CC3M-595K, followed by instruction tuning with MIMIC-IT,  LLaVA and Video-ChatGPT.

**Strengths:**

- The proposed multi-scale feature and dynamic visual tokens are interesting and effective.
- In my humble opinion, this work could be one of the first to explore multimodal in-context instruction tuning in the video domain.
- Strong results on multiple benchmarks. Detailed analysis with GPT evaluator.
- The paper is well-written and easy to follow.

**Weaknesses:**

- Gathering high-quality video instruction data remains very challenging when aiming for large-scale training. This paper requires very high-quality video instruction data to learn well. In other words, it seems difficult to further scale-up the training for this approach.
- This method seems not able to process long videos or movies.
- The model is built upon LLMs which outputs text tokens only. As the community is moving forward to any-to-any large multimodal models, the proposed approach seems less promising to extend toward that direction.
- It remains unclear whether the model will generate hallucinations in the temporal dimension.

**Questions:**

see weakness

---

### Official Review · Reviewer_4Vet · 2023-11-01

**Soundness:** 2 fair
**Presentation:** 3 good
**Contribution:** 1 poor
**Rating:** 3
**Confidence:** 5

**Summary:**

This paper proposed a unified vision-language model combined with LLM for image and video understanding. The authors adopt an efficient token-merging method for encoding images and videos with variance length. They jointly train a model on both image and video instruction tuning data. The model achieves good results on several image and video understanding tasks. They conduct extensive ablations and analyses to verify the effectiveness of their methods.

**Strengths:**

- It is interesting to develop a unified and efficient LLM-based vision-language model.
- Token merging is an efficient method for reducing the number of tokens from video, especially suitable for LLM-based vision-language models.
- The experiments are extensive. The evaluations and analyses are sufficient. The overall writing is clear.

**Weaknesses:**

- The novelty is somewhat limited. Token merging is an existing method proposed by Bolya et al. [R1] nearly one year ago and is very similar to the token merging used in this paper. The authors have not included it in the related works.
- The idea of training a unified video-image-language model is not novel either. Some works have widely explored training a unified video-image-language model, although they are not LLM-based. They should be carefully described in the related works. [R2, R3]
- The performance comparisons are unfair because their model is based on LLaMA-2, while other baselines are based on LLaMA-1. Their model also use more data than LLaVA-13B.
- The details of the single-scale are missed, are the tokens from step 1? The number of tokens of single-scale and multi-scale is not the same, would this cause the performance difference?
- There are no experiment results to support the conclusion that other methods do not benefit from joint training on image and video data in Table A.
- How about the performance of increasing the query tokens in Q-former-based models to 112 (the same as used in their model), may be it will be a strong baseline.
-

[R1] Token Merging: Your ViT But Faster, ICLR 2023.

[R2] Frozen in Time: A Joint Video and Image Encoder for End-to-End Retrieval, ICCV 2021.

[R3] OmniVL: One Foundation Model for Image-Language and Video-Language Tasks, NeurIPS 2022.

**Questions:**

Refering to the weaknesses.